# Real-World Image Variation by Aligning Diffusion Inversion Chain

Yuechen Zhang[1]    Jinbo Xing[1]    Eric Lo[1]    Jiaya Jia[1,2]

[1]The Chinese University of Hong Kong    [2]SmartMore

{yczhang21, jbxing, ericlo, leojia}@cse.cuhk.edu.hk

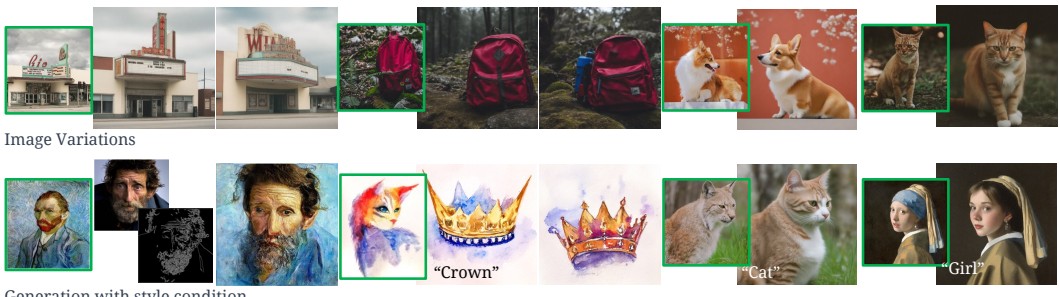

Image Variations

Generation with style condition

"Crown"                          "Cat"                    "Girl"

Figure 1: Leveraging an image exemplar, our training-free method excels in image generation tasks such as generating (a) image variations, (b) images of a similar style (with conditions).

## Abstract

Recent diffusion model advancements have enabled high-fidelity images to be generated using text prompts. However, a domain gap exists between generated images and real-world images, which poses a challenge in generating high-quality variations of real-world images. Our investigation uncovers that this domain gap originates from a latents' distribution gap in different diffusion processes. To address this issue, we propose a novel inference pipeline called **R**eal-world **I**mage **V**ariation by **AL**ignment (RIVAL) that utilizes diffusion models to generate image variations from a single image exemplar. Our pipeline enhances the generation quality of image variations by aligning the image generation process to the source image's inversion chain. Specifically, we demonstrate that step-wise latent distribution alignment is essential for generating high-quality variations. To attain this, we design a cross-image self-attention injection for feature interaction and a step-wise distribution normalization to align the latent features. Incorporating these alignment processes into a diffusion model allows RIVAL to generate high-quality image variations without further parameter optimization. Our experimental results demonstrate that our proposed approach outperforms existing methods concerning semantic similarity and perceptual quality. This generalized inference pipeline can be easily applied to other diffusion-based generation tasks, such as image-conditioned text-to-image generation and stylization. Project page: https://rival-diff.github.io

## 1 Introduction

Generating real-world image variation is a crucial area of research in computer vision and machine learning, owing to its practical applications in image editing, synthesis, and data augmentation [1, 2]. This task involves the generation of diverse variations of a given real-world image while preserving its semantic content and visual quality. Early methods for generating image variations included texture

37th Conference on Neural Information Processing Systems (NeurIPS 2023).

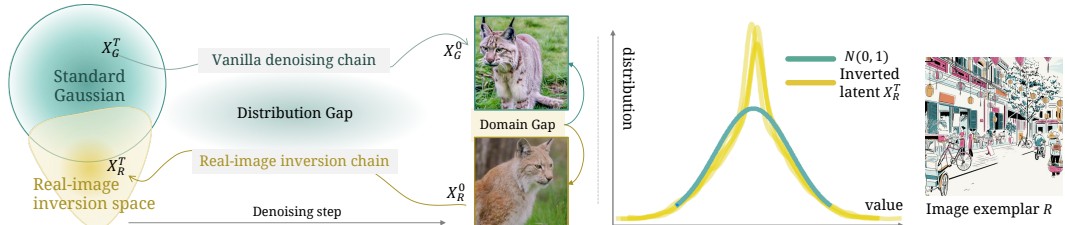

Figure 2: Conceptual illustration of the challenges in the real-world image variation. Left: Despite using the same text prompt `"a lynx sitting in the grass"`, a bias exists between the latent distribution in the vanilla generation chain and the image inversion chain, resulting in a significant domain gap between the denoised images. Right: Visualization of the real-image inverted four-channel latent. The distribution of the latent is biased in comparison to a standard Gaussian distribution.

synthesis, neural style transfer, and generative models [3, 4, 5, 6, 7, 8]. However, these methods are limited in generating realistic and diverse variations from real-world images and are only suited for generating variations of textures or artistic images.

Denoising Diffusion Probabilistic Models (DDPMs) have resulted in significant progress in text-driven image generation [9, 10, 11]. However, generating images that maintain the style and semantic content of the reference remains a significant challenge. Although advanced training-based methods [12, 13, 14, 15] can generate images with novel concepts and styles with given images, they require additional training stages and data. Directly incorporating image as the input condition [11, 16] results in suboptimal visual quality and content diversity compared to the reference input. Besides, they do not support input with text descriptions. A plug-and-play method has not been proposed to generate high-quality, real-world image variations without extra optimization.

Observing the powerful denoising ability of pre-trained DDPMs in recovering the original image from the inverted latent space [17, 18], we aim to overcome the primary challenge by modifying the vanilla latent denoising chain of the DDPM to fit the real-image inversion chain [17, 18, 19]. Despite generating images with the same text condition, a significant domain gap persists between the generated and source images, as depicted in Fig. 2. We identify distribution misalignment as the primary factor that impedes the diffusion model from capturing certain image features from latents in the inversion chain. As illustrated in the right portion of Fig. 2, an inverted latent may differ significantly from the standard Gaussian distribution. This misalignment accumulates during the denoising process, resulting in a domain gap between the generated image and its reference exemplar.

To address this distribution gap problem for generating image variations, we propose an pure inference pipeline called Real-world Image Variation by Alignment (RIVAL). RIVAL is a tunning-free approach that reduces the domain gap between the generated and real-world images by aligning the denoising chain with the real-image inversion chain. Our method comprises two key components: (i) a cross-image self-attention injection that enables cross-image feature interaction in the variation denoising chain, guided by the hidden states from the inversion chain, and (ii) a step-wise latent normalization that aligns the latent distribution with the inverted latent in early denoising steps. Notably, this modified inference process requires no training and is suitable for arbitrary image input.

As shown in Fig. 1, our proposed approach produces visually appealing image variations while maintaining semantic and style consistency with a given image exemplar. RIVAL remarkably improves the quality of image variation generation qualitatively and quantitatively compared to existing methods [11, 16, 20]. Furthermore, we have demonstrated that RIVAL's alignment process can be applied to other text-to-image tasks, such as text-driven image generation with real-image condition [21, 22, 23] and example-based inpainting [24, 25].

This paper makes three main contributions. (1) Using a real-world image exemplar, we propose a novel tunning-free approach to generate high-quality image variations. (2) We introduce an latent alignment process to enhance the quality of the generated variations. (3) Our proposed method offers a promising denoising pipeline that can be applied across various applications.

## 2 Related Works

**Diffusion models with text control** represent advanced techniques for controllable image generation with text prompts. With the increasing popularity of text-to-image diffusion models [11, 16, 26, 10,

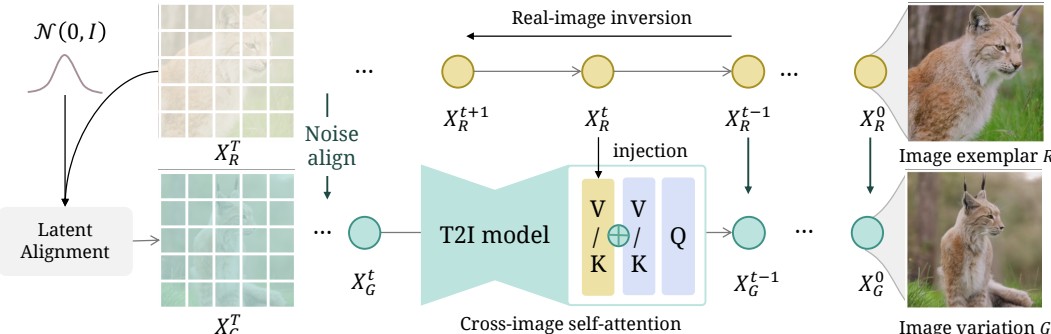

Figure 3: High-level framework of RIVAL. Input exemplar $R$ is inverted to a noisy latent $X_R^T$. An image variation $G$ is generated from random noise following the same distribution as $X_R^T$. For each denoising step $t$, we interact $X_R^t$ and $X_G^t$ by self-attention injection and latent alignment.

27], high-quality 2D images can be generated through text prompt input. However, such methods cannot guarantee to generate images with the same low-level textures and tone mapping as the given image reference, which is difficult to describe using text prompts. Another line of works aims for diffusion-based concept customization, which requires fine-tuning the model and transferring image semantic contents or styles into text space for new concept learning [12, 13, 14, 15, 28]. However, this involves training images and extra tuning, making it unsuitable for plug-and-play inference.

**Real-world image inversion** is a commonly employed technique in Generative Adversarial Networks (GANs) for image editing and attribute manipulation [6, 29, 30, 31, 32]. It involves inverting images to the latent space for reconstructions. In diffusion models, the DDIM sampling technique [17] provides a deterministic and approximated invertible diffusion process. Recently developed inversion methods [18, 19] guarantee high-quality reconstruction with step-wise latent alignments. With diffusion inversion, real-image text-driven manipulations can be performed in image and video editing methods [33, 34, 23, 22, 35, 36, 37]. These editing methods heavily rely on the original structure of the input, thus cannot generate free-form variations with the same content as the reference image from the perspective of image generation.

**Image variation generation** involves generating diverse variations of a given image exemplar while preserving its semantic content and visual quality. Several methods have been proposed for this problem, including neural style transfer [5, 38, 39, 40], novel view synthesis [41, 42, 43], and GAN-based methods [7, 44, 32]. However, these methods were limited to generating variations of artistic images or structure-preserved ones and were unsuitable for generating realistic and diverse variations of real-world images. Diffusion-based methods [16, 11, 20] can generate inconsistent image variants by analyzing the semantic contents from the reference while lacking low-frequency details. Recent methods using image prompt to generate similar contents or styles [45, 46, 47, 48] requires additional case-wise tuning or training. Another concurrent approach, MasaCtrl [23], adopts self-attention injection as guidance for the denoising process. Yet, it can only generate variations with a generated image and fails on real-world image inputs. In contrast, RIVAL leverages the strengths of diffusion chain alignment to generate variations of real-world images.

## 3 Real-World Image Variation by Alignment

In this work, we define the image variation as the construction of a diffusion process $F$ that satisfies $D(R, C) \approx D(F_C(X, R), C)$, where $D(\cdot, C)$ is a data distribution function with a given semantic content condition $C$. The diffusion process $F_C(X, R) = G$ generates the image variation $G$ based on a sampled latent feature $X$, condition $C$, and the exemplar image $R$.

The framework of RIVAL is illustrated in Fig. 3. RIVAL generates an inverted latent feature chain $\{X_R^0, ..., X_R^T\}$ by inverting a reference exemplar image $R$ using DDIM inversion [17]. Then we obtain the initial latent $X_R^T$ of the inversion chain. Next, a random latent feature $X_G^T$ is sampled as the initial latent of the image generation (denoising) chain. In this multi-step denoising chain $\{X_G^T, ..., X_G^0\}$ for generation, we align the latent features to latents from the inversion chain to obtain perceptually similar generations. The modified step-wise denoising function $f$ at step $t$ can be represented abstractly as:

$$X_G^t = f_t(X_G^{t+1}, X_R^{t+1}, C). \tag{1}$$

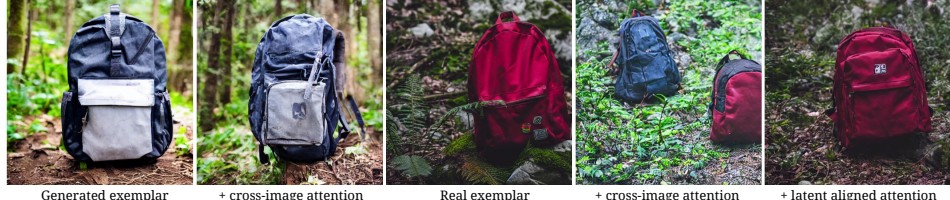

| Generated exemplar | + cross-image attention | Real exemplar | + cross-image attention | + latent aligned attention |

Figure 4: Exemplar images and generation results are obtained using the prompt `"backpack in the wild"`. Cross-image self-attention can generate faithful outputs when a vanilla generation chain is employed as guidance. In the case of real-world images, the generation of faithful variations is dependent on the alignment of the latent.

This function is achieved by performing adaptive cross-image attention in self-attention blocks for feature interaction and performing latent distribution alignment in the denoising steps. The denoising chain can produce similar variations by leveraging latents in the inversion chain as guidance.

## 3.1 Cross-Image Self-Attention Injection

To generate image variations from an exemplar image $R$, a feature interaction between the inversion chain and the generation chain is crucial. Previous works [49, 22] have shown that self-attention can efficiently facilitate feature interactions. Similar to its applications [33, 23, 37] in text-driven generation and editing tasks, we utilize intermediate hidden states $\mathbf{v}_R$ obtained from the inversion chain to modify the self-attention in the generation chain. Specifically, while keeping the inversion chain unchanged, our modified Key-Value features for cross-image self-attention for one denoising step $t$ in the generation chain are defined as follows:

$$Q = W^Q(\mathbf{v}_G), K = W^K(\mathbf{v}'_G), V = W^V(\mathbf{v}'_G), \text{ where} \tag{2}$$

$$\mathbf{v}'_G = \begin{cases} \mathbf{v}_G \oplus \mathbf{v}_R & \text{if } t \leq t_{\text{align}} \\ \mathbf{v}_R & \text{otherwise} \end{cases}. \tag{3}$$

We denote $W^{(\cdot)}$ as the pre-trained, frozen projections in the self-attention block and use $\oplus$ to represent concatenation in the spatial dimension. Specifically, we adopt an **Attention Fusion** strategy. In early steps ($t > t_{\text{align}}$), we replace the KV values with $W^V(\mathbf{v}_R)$, $W^K(\mathbf{v}_R)$ using the hidden state $\mathbf{v}_R$ from the inversion chain. In subsequent steps, we concatenate $\mathbf{v}_R$ and the hidden state from the generation chain itself $\mathbf{v}_G$ to obtain new Key-Values. We do not change Q values and maintain them as $W^Q(\mathbf{v}_G)$. The proposed adaptive cross-image attention mechanism explicitly introduces feature interactions in the denoising process of the generation chain. Moreover, Attention Fusion strategy aligns the content distribution of the latent features in two denoising chains.

Building upon the self-attention mechanism [50], we obtain the updated hidden state output as $\mathbf{v}^*_G = \text{softmax}\left(QK^\top / \sqrt{d_k}\right)V$, where $d_k$ is the dimensionality of $K$. Cross-image self-attention and feature injection can facilitate the interaction between hidden features in the generation chain and the inversion chain. It should be noted that the inversion chain is deterministic throughout the inference process, which results in reference feature $\mathbf{v}_R$ remaining independent of the generation chain.

## 3.2 Inverted Latent Chain Alignment

The cross-image self-attention injection is a potent method for generating image variations from a vanilla denoising process originating from a standard Gaussian distribution and is demonstrated in [23]. However, as depicted in Fig. 4, direct adaptation to real-world image input is not feasible due to a domain gap between real-world inverted latent chains and the vanilla generation chains. This leads to the attenuation of attention correlation during the calculation of self-attention. We further visualize this problem in Fig. 12. To facilitate the generation of real-world image variations, the pseudo-generation chain (inversion chain) of the reference exemplar can be estimated using the DDIM inversion [17]. Generating an image from latent features $X^T$ using a small number of denoising steps is possible with the use of deterministic DDIM sampling:

$$X^{t-1} = \sqrt{\alpha_{t-1}/\alpha_t} \cdot X^t + \sqrt{\alpha_{t-1}}(\beta_{t-1} - \beta_t) \cdot \varepsilon_\theta(X^t, t, \mathcal{C}), \tag{4}$$

where step-wise coefficient is set to $\beta_t = \sqrt{1/\alpha_t - 1}$ and $\varepsilon_\theta(X^t, t, \mathcal{C})$ is the pre-trained noise prediction function in one timestep. Please refer to Appendix A for the details of $\alpha_t$ and DDIM

sampling. With the assumption that noise predictions are similar in latents with adjacent time steps, DDIM sampling can be reversed In a small number of steps $t \in [0, T]$, using the equation:

$$X^{t+1} = \sqrt{\alpha_{t+1}/\alpha_t} \cdot X^t + \sqrt{\alpha_{t+1}}(\beta_{t+1} - \beta_t) \cdot \varepsilon_\theta\big(X^t, t, \mathcal{C}\big), \tag{5}$$

which is known as DDIM inversion [17]. We apply DDIM to acquire the latent representation $X_R^T$ of a real image $R$. Nevertheless, the deterministic noise predictions in DDIM cause uncertainty in maintaining a normal distribution $\mathcal{N}(\mathbf{0}, \boldsymbol{I})$ of the inverted latent $X_R^T$. This deviation from the target distribution causes the attention-attenuation problem [51] between generation chains of $R$ and $G$, leading to the generation of images that diverge from the reference. To tackle this challenge, we find a straightforward distribution alignment on the initial latent $X_G^T$ useful, that is,

$$x_G^T \in X_G^T \sim \mathcal{N}\big(\mu\big(X_R^T\big), \sigma^2\big(X_R^T\big)\big), \text{ or } X_G^T = \text{shuffle}\big(X_R^T\big). \tag{6}$$

In the alignment process, pixel-wise feature elements $x_G^T \in X_G^T$ in the spatial dimension can be initialized from one of the two sources: (1) an adaptively normalized Gaussian distribution or (2) permutated samples from the inverted reference latent, $\text{shuffle}(X_R^T)$. Our experiments show that both types of alignment yield comparable performance. This latent alignment strengthens the association between the variation latents and the exemplar's latents, increasing relevance within the self-attention mechanism and thus increasing the efficacy of the proposed cross-image attention.

In the context of initializing the latent variable $X_G^T$, further alignment steps within the denoising chain are indispensable. This necessity arises due to the limitations of the Classifier-Free Guidance (CFG) inference proposed in [52]. While effective for improving text prompt guidance in the generation chain, CFG cannot be directly utilized to reconstruct high-quality images through DDIM inversion. On the other hand, the noise scaling of CFG will affect the noise distribution and lead to a shift between latents in two chains, as shown in Fig. 11. To avoid this misalignment while attaining the advantage of the text guidance in CFG, we decouple two inference chains and rescale the noise prediction during denoising inference, formulated as an adaptive normalization [38]:

$$\epsilon_\theta\big(X_G^t, t, \mathcal{C}\big) = \text{AdaIN}(\varepsilon_\theta^{\text{cfg}}\big(X_G^t, t, \mathcal{C}\big), \varepsilon_\theta\big(X_R^t, t, \mathcal{C}\big)), \text{ where} \tag{7}$$

$$\varepsilon_\theta^{\text{cfg}}\big(X_G^t, t, \mathcal{C}\big) = m\varepsilon_\theta\big(X_G^t, t, \mathcal{C}\big) + (1 - m)\varepsilon_\theta\big(X_G^t, t, \mathcal{C}^*\big), \text{ and } t > t_{\text{early}}. \tag{8}$$

$\varepsilon_\theta^{\text{cfg}}(X_G^t, t, \mathcal{C})$ is the noise prediction using guidance scale $m$ and unconditional null-text $C^*$. We incorporate this noise alignment approach during the initial stages ($t > t_{\text{early}}$). After that, a vanilla CFG scaling is applied as the content distribution in $G$ may not be consistent with $R$. Aligning the distributions at this early stage proves advantageous for aligning the overall denoising process.

## 4 Experiments

We evaluate our proposed pipeline through multiple tasks in Sec. 4.2: image variation, image generation with text and image conditions, and example-based inpainting. Additionally, we provide quantitative assessments and user study results to evaluate generation quality in Sec. 4.3. A series of ablation studies and discussions of interpretability are conducted in Sec. 4.4 to assess the effectiveness of individual modules within RIVAL.

### 4.1 Implementation Details

Our study obtained a high-quality test set of reference images from the Internet and DreamBooth [12] to ensure a diverse image dataset. To generate corresponding text prompts $C$, we utilized BLIP2 [54]. Our baseline model is Stable-Diffusion V1.5. During the image inversion and generation, we employed DDIM sample steps $T = 50$ for each image and set the classifier-free guidance scale $m = 7$ in Eq. (8). We split two stages at $t_{\text{align}} = t_{\text{early}} = 30$ for attention alignment in Eq. (3) and latent alignment in Eq. (8). In addition, we employ the shuffle strategy described in Eq. (6) to initialize the starting latent $X_G^T$. Experiments run on a single NVIDIA RTX4090 GPU with 8 seconds to generate image variation with batch size 1.

### 4.2 Applications and Comparisons

**Image variation.** As depicted in Fig. 5, real-world image variations exhibit diverse characteristics to evaluate perceptual quality. Text-driven image generation using the basic Stable Diffusion [11] cannot use image exemplars explicitly, thus failing to get faithful image variations. While recent image-conditioned generators such as [20, 53, 16] have achieved significant progress for image input,

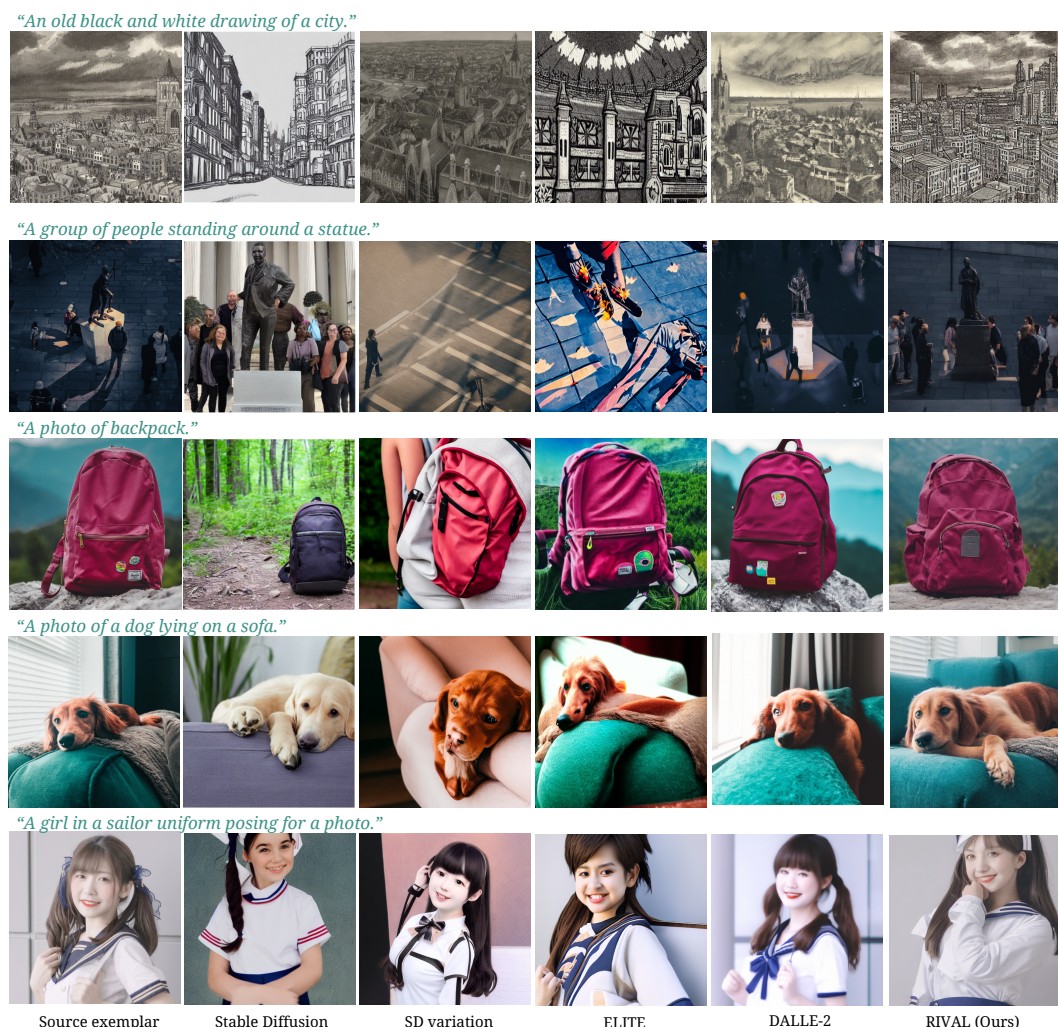

*"An old black and white drawing of a city."*

*"A group of people standing around a statue."*

*"A photo of backpack."*

*"A photo of a dog lying on a sofa."*

*"A girl in a sailor uniform posing for a photo."*

| Source exemplar | Stable Diffusion | SD variation | ELITE | DALLE-2 | RIVAL (Ours) |

Figure 5: Real-world image variation. We compare RIVAL with recent competitive methods [11, 53, 20] conditioned on the same text prompt (the 2nd column) or exemplar image (3rd-5th columns).

their reliance on encoding images into text space limits the representation of certain image features, including texture, color tone, lighting environment, and image style. In contrast, RIVAL generates image variations based on text descriptions and reference images, harnessing the real-image inversion chain to facilitate latent distribution alignments. Consequently, RIVAL ensures high visual similarity in both semantic content and low-level features. For instance, our approach is the sole technique that generates images with exceedingly bright or dark tones (2nd-row in Fig. 5). More visual comparisons and experimental settings can be found in Appendix C.

**Text-driven image generation.** In addition to the ability to generate images corresponding to the exemplar image and text prompts, we have also discovered that RIVAL has a strong ability to transfer styles and semantic concepts in the exemplar for a casual text-driven image generation. When an inversion with the original text prompt is performed, the alignment on denoising processes can still transfer high-level reference style to the generated images with modified prompts. This process could be directly used in structure-preserved editing. This could be regarded as the same task of MasaCtrl and Plug-and-Play [23, 34], which utilizes an initialized inverted latent representation to drive text-guided image editing. A comparison is shown in the right part of Fig. 6, here we directly use the inverted latent $X_G^T = X_R^T$ for two chains. We adopt the same interaction starting step $t = 45$ as the MasaCtrl.

Furthermore, in the absence of a structure-preserving prior, it is possible for a user-defined text prompt to govern the semantic content of a freely generated image $G$. This novel capability is showcased

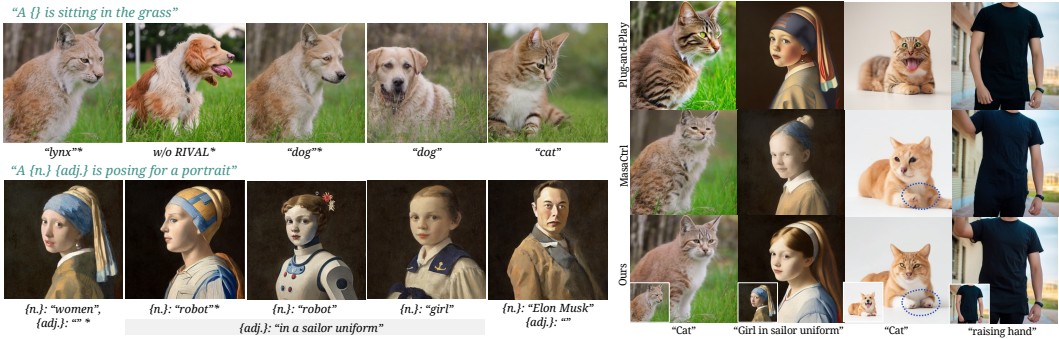

Figure 6: Left: Text-driven image generation results using RIVAL, with the leftmost image as the source exemplar. RIVAL can generate images with diverse text inputs while preserving the reference style. Images noted with * indicate structure-preserved editing, which starts with the same initialized inverted reference latent $X_R^T$ as the exemplar. Right: a visual comparison of structure-preserved image editing with recent approaches [34, 23].

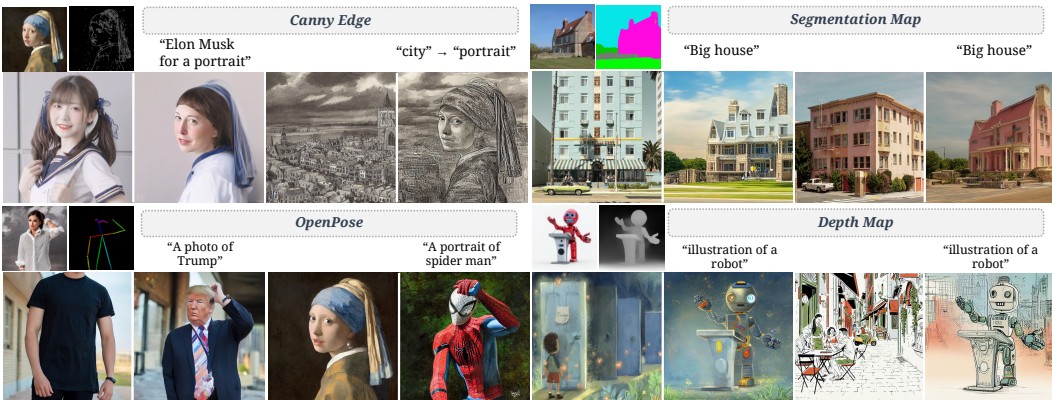

Figure 7: The availability of RIVAL with ControlNet [55]. Two examples are given for each modality of the control condition. Exemplars are shown on the left of each image pair.

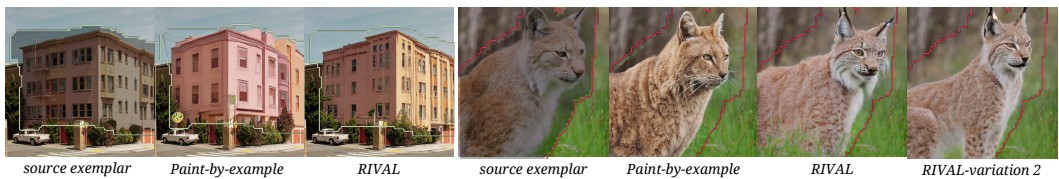

Figure 8: RIVAL extended to self-example image inpainting. We use the same image as the example to fill the masked area. Results are compared with a well-trained inpainting SOTA method [25].

in the left part of Fig. 6, where RIVAL successfully generates images driven by text input while maintaining the original style of the reference image. Further, to extend this ability to other image conditions, we adapt RIVAL with ControlNet Fig. 7 by adding condition residuals of ControlNet blocks on the generation chain, as shown in Fig. 7. With RIVAL, we can easily get a style-specific text-to-image generation. For instance, it can produce a portrait painting of a robot adorned in a sailor uniform while faithfully preserving the stylistic characteristics.

**Example-based image inpainting.** When abstracting RIVAL as a novel paradigm of image-based diffusion inference, we can extend this framework to enable it to encompass other image editing tasks like inpainting. Specifically, by incorporating a coarse mask $M$ into the generation chain, we only permute the inverted latent variables within the mask to initialize the latent variable $X_G^T$. During denoising, we replace latents in the unmasked regions with latents in the inversion chain for each step. As shown in Fig. 8, we present a visual comparison between our approach and [25]. RIVAL produces a reasonable and visually harmonious inpainted result within the mask.

| Metric | SD [11] | ImgVar [53] | ELITE [20] | DALL·E 2 [16] | **RIVAL** |
|---|---|---|---|---|---|
| Text Alignment ↑ | $\underline{0.255} \pm 0.04$ | $0.223 \pm 0.04$ | $0.209 \pm 0.05$ | $0.253 \pm 0.04$ | $\mathbf{0.275} \pm 0.03$ |
| Image Alignment ↑ | $0.748 \pm 0.08$ | $0.832 \pm 0.07$ | $0.736 \pm 0.09$ | $\mathbf{0.897} \pm 0.05$ | $\underline{0.840} \pm 0.07$ |
| Palette Distance ↓ | $3.650 \pm 1.30$ | $3.005 \pm 0.86$ | $2.885 \pm 0.81$ | $\underline{2.102} \pm 0.83$ | $\mathbf{1.674} \pm 0.63$ |
| Real-world Authenticity ↓ | - | 2.982 (4.7%) | 3.526 (9.6%) | 1.961 (28.0%) | **1.530** (61.6%) |
| Condition Adherence ↓ | - | 3.146 (7.3%) | 3.353 (5.6%) | 1.897 (30.6%) | **1.603** (56.4%) |

Table 1: Quantitative comparisons. We evaluate the quality of image variation regarding feature matching within different levels (color palette, text feature, image feature), highlighted with **best** and second best results. We also report user preference rankings and the first ranked rate.

| Methods | Text Align. ↑ | Image Align. ↑ | Palette Dist. ↓ | LPIPS ↓ | Preparation Time (s) ↓ | Inference Time (s) ↓ |
|---|---|---|---|---|---|---|
| PnP [34] | **0.249** | 0.786 | 1.803 | **0.245** | 200 | 21 |
| MasaCtrl [23] | 0.226 | 0.827 | 1.308 | 0.274 | 6 | 15 |
| **RIVAL** | 0.231 | **0.831** | **1.192** | **0.245** | 6 | 15 |

Table 2: Quantitative comparisons with image editing methods. Both in performance and efficiency (time consumption on one NVIDIA RTX3090 for fair comparison).

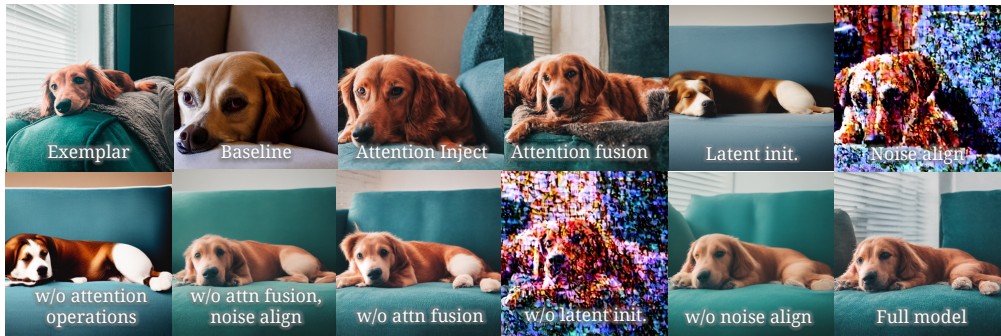

Figure 9: Left: a visual example for a module-wise ablation study.

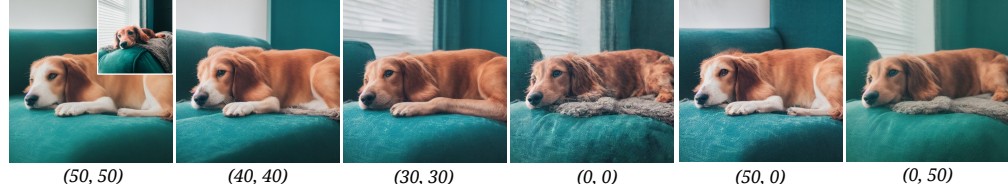

Figure 10: Ablation study for different early-alignment strategies with the exemplar in Fig. 5. We list $(t_{\text{align}}, t_{\text{early}})$ pairs for each image. All images are generated from the same fixed latent.

## 4.3 Quantitative Evaluation

We compare RIVAL with several state-of-the-art methods, employing the widely used CLIP-score [56] evaluation metric for alignments with text and the image exemplar. The evaluation samples are from two datasets: the DreamBooth dataset [12] and our collected images. In addition, we cluster a 10-color palette using $k$-means and calculate the minimum bipartite distance to assess the similarity in low-level color tones. In general, the results in Tab. 1 demonstrate that RIVAL significantly outperforms other methods regarding semantic and low-level feature matching. Notably, our method achieves better text alignment with the real-image condition than the vanilla text-to-image method using the same model. Furthermore, our alignment results are on par with those attained by DALL·E 2 [16], which directly utilizes the image CLIP feature as the condition.

We conducted a user study to evaluate further the generated images' perceptual quality. This study utilized results generated by four shuffled methods based on the same source. Besides, the user study

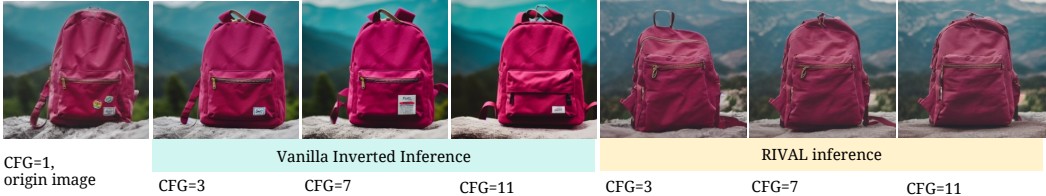

CFG=1,
origin image

Vanilla Inverted Inference

RIVAL inference

CFG=3          CFG=7          CFG=11          CFG=3          CFG=7          CFG=11

Figure 11: High CFG artifacts defense. Left: vanilla generation, starting from the DDIM inverted latent. Right: RIVAL generated images, starting from the permuted latents.

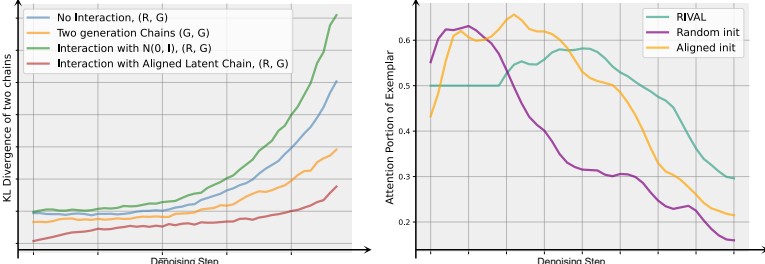

Figure 12: RIVAL's interpretability. Left: KL divergence between latent among two chains across denoising steps. Right: attention score with the reference feature with different alignment strategies.

includes a test for image authenticity, which lets users identify the "real" image among generated results. We collected responses of ranking results for each case from 41 participants regarding visual quality. The findings indicate a clear preference among human evaluators for our proposed approach over existing methods [53, 20, 16]. The detailed ranking results are presented in Tab. 1.

We also compare RIVAL with PnP and MasaCtrl for image editing on a representative test set, with additional metrics in Perceptual Similarity [57] and inference time. Our method presents a competitive result in Tab. 2.

## 4.4 Ablation Studies

We perform ablation studies to evaluate the efficacy of each module in our design. Additional visual and quantitative results can be found in Appendix D.

**Cross-image self-attention injection** plays a crucial role in aligning the feature space of the source image and the generated image. We eliminated the cross-image self-attention module and generated image variations to verify this. The results in Fig. 9 demonstrate that RIVAL fails to align the feature space without attention injection, generating images that do not correspond to the exemplar.

**Latent alignment.** To investigate the impact of step-wise latent distribution alignment, we remove latent alignments and sampling $X_G^T \sim \mathcal{N}(\mathbf{0}, \mathbf{I})$. An example is shown in Fig. 4 and Fig. 9. Without latent alignment, RIVAL produces inconsistent variations of images, exhibiting biased tone and semantics compared to the source image. Additionally, we perform ablations on attention fusion in Eq. (3) and noise alignment in Eq. (8) during the early steps. Clear color and semantic bias emerge when the early alignment is absent, as shown in Fig. 10. Furthermore, if the values of $t_{\text{early}}$ or $t_{\text{align}}$ are excessively small, the latent alignment stage becomes protracted, resulting in the problem of over-alignment. Over-alignment gives rise to unwanted artifacts and generates blurry outputs.

**Interpretability.** We add experiments for the interpretability of RIVAL in the following two aspects. *(a) Latent Similarity*. To assess RIVAL components' efficacy regarding the distribution gap, we illustrate the KL divergence of noisy latent between chains A and B, $X_A^t$ and $X_B^t$ in the generation process, as depicted in Fig. 12 left part. Interactions between different distributions (green) widen the gap, while two generation chains with the same distribution (orange) can get a better alignment by attention interactions. With aligned latent chain alignment and interaction, RIVAL (purple) effectively generates real-world image variations.

*(b) Reference Feature Contribution*. Attention can be viewed as sampling value features from the key-query attention matrix. RIVAL converts self-attention to image-wise cross-attention. When

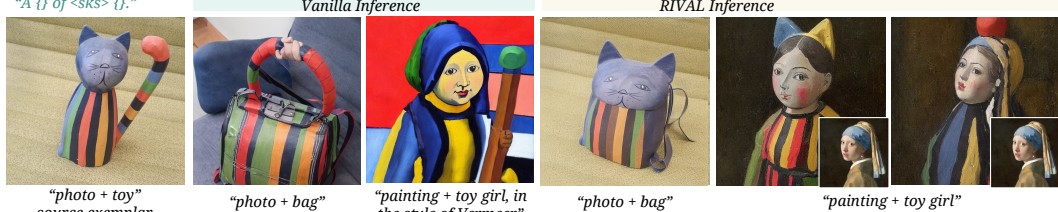

*"A {} of <sks> {}."*      *Vanilla Inference*      *RIVAL Inference*

*"photo + toy"*
*source exemplar*    *"photo + bag"*    *"painting + toy girl, in the style of Vermeer"*    *"photo + bag"*    *"painting + toy girl"*

Figure 13: RIVAL enables seamless customization of optimized novel concepts through text prompt control (`<sks>`). With various text prompt inputs, we can still generate images preserving the tones, style, and contents of the provided source exemplar (depicted on the left).

latents are sourcing from the same distribution ($X_G^T, X_R^T \sim \mathcal{N}(0, I)$) images retain consistent style and content attributes (as in Fig. 12 left orange). This result is beneficial since we do not require complex text conditions $c$ to constrain generated images to get similar content and style. For a more direct explanation, we visualize the bottleneck feature contributions of attention score presented in Fig. 12 right part. Reference contribution of the softmax score is denoted as score$_R = \sum_{v_i \in \mathbf{v}_R}(W^Q\mathbf{v}_G \cdot (W^K(v_i))^\top)/\sum_{v_j \in \mathbf{v}_G \oplus \mathbf{v}_R}(W^Q\mathbf{v}_G \cdot (W^K(v_j))^\top)$. As RIVAL adopts early fusion in the early steps, we use 50% as the score in the early steps. Latent initialization (orange) and with early alignments (green) play critical roles in ensuring a substantial contribution of the source feature in self-attention. A higher attention contribution helps in resolving the attention attenuation problem (purple) in the generation process.

## 4.5 Discussions

**Integration with concept customization.** In addition to its ability to generate image variations from a single source image using a text prompt input for semantic alignment, RIVAL can be effectively combined with optimization-based concept customization techniques, such as DreamBooth [12], to enable novel concept customization. As illustrated in Fig. 13, an optimized concept can efficiently explore the space of potential image variations by leveraging RIVAL's proficiency in real-world image inversion alignment.

**Limitations and Future Directions.** Despite introducing an innovative technique for crafting high-quality inconsistent image variations, our method is contingent upon a text prompt input, potentially infusing semantic biases affecting image quality. As evidenced in Fig. 14, a prompt like "Pokemon" may lean towards popular choices like "Pikachu" due to training set biases, resulting in a Pikachu-dominated generation. Besides, the base model struggles to generate complex scenes and complicated concepts, e.g., "illustration of a little boy standing in front of a list of doors with butterflies around them in Fig. 14 (c)". This complexity can degrade the inversion chain and widen the domain gap, leading to less accurate results. Future studies might concentrate on refining diffusion models and exploring novel input avenues besides text prompts to mitigate these constraints.

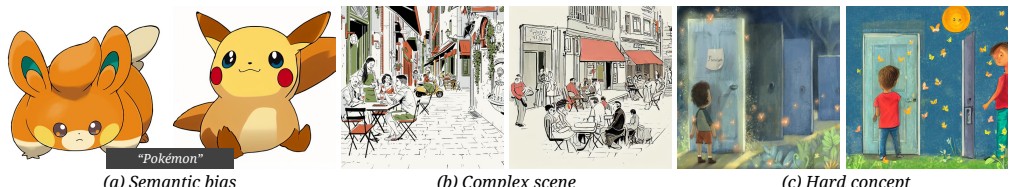

*(a) Semantic bias*      *(b) Complex scene*      *(c) Hard concept*

Figure 14: Fail cases of RIVAL. The exemplar image is on the left for each case.

## 5 Conclusion

This paper presents a novel pipeline for generating diverse and high-quality variations of real-world images while maintaining their semantic content and style. Our proposed approach addresses previous limitations by modifying the diffusion model's denoising inference chain to align with the real-image inversion chain. We introduce a cross-image self-attention injection and a step-wise latent alignment technique to facilitate alignment between two chains. Our method exhibits significant improvements in the quality of image variation generation compared to state-of-the-art methods, as demonstrated through qualitative and quantitative evaluations. Moreover, with the novel paradigm of hybrid text-image conditions, our approach can be easily extended to multiple text-to-image tasks.

**Acknowledgements.** This work is partially supported by the Research Grants Council under the Areas of Excellence scheme grant AoE/E-601/22-R, Hong Kong General Research Fund (14208023), Hong Kong AoE/P-404/18, and Centre for Perceptual and Interactive Intelligence (CPII) Limited under the Innovation and Technology Fund. We are grateful to Jingtao Zhou for the meaningful discussions.

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

# Appendix

## A  Basic Background of Diffusion Models

This section uses a modified background description provided in [18]. We only consider the condition-free case for the diffusion model here. Diffusion Denoising Probabilistic Models (DDPMs) [58] are generative latent variable models designed to approximate the data distribution $q(x_0)$. The diffusion operation starts from the latent $x_0$, adding step-wise noise to diffuse data into pure noise $x_T$. It's important to note that this process can be viewed as a Markov chain starting from $x_0$, where noise is gradually added to the data to generate the latent variables $x_1, \ldots, x_T \in X$. The sequence of latent variables follows the conditional distribution $q(x_1, \ldots, x_t \mid x_0) = \prod_{i=1}^{t} q(x_t \mid x_{t-1})$. Each step in the forward process is defined by a Gaussian transition $q(x_t \mid x_{t-1}) := \mathcal{N}(x_t; \sqrt{1 - k_t}x_{t-1}, k_t I)$, which is parameterized by a schedule $k_0, \ldots, k_T \in (0, 1)$. As $T$ becomes sufficiently large, the final noise vector $x_T$ approximates an isotropic Gaussian distribution.

The forward process allows us to express the latent variable $x_t$ directly as a linear combination of noise and $x_0$, without the need to sample intermediate latent vectors.

$$x_t = \sqrt{\alpha_t}x_0 + \sqrt{1 - \alpha_t}w, \quad w \sim \mathcal{N}(\mathbf{0}, \boldsymbol{I}), \tag{9}$$

where $\alpha_t := \prod_{i=1}^{t}(1 - k_i)$. To sample from the distribution $q(x_0)$, a reversed denoising process is defined by sampling the posteriors $q(x_{t-1} \mid x_t)$, which connects isotropic Gaussian noise $x_T$ to the actual data. However, the reverse process is computationally challenging due to its dependence on the unknown data distribution $q(x_0)$. To overcome this obstacle, an approximation of the reverse process with a parameterized Gaussian transition network denoted as $p_\theta(x_{t-1} \mid x_t)$, where $p_\theta(x_{t-1} \mid x_t)$ follows a normal distribution with mean $\mu_\theta(x_t, t)$ and covariance $\Sigma_\theta(x_t, t)$. As an alternative approach, the prediction of the noise $\epsilon_\theta(x_t, t)$ added to $x_0$, which is obtained using equation 9, can replace the use of $\mu_\theta(x_t, t)$ as suggested in [58]. Bayes' theorem could be applied to approximate

$$\mu_\theta(x_t, t) = \frac{1}{\sqrt{\alpha_t}}\left(x_t - \frac{k_t}{\sqrt{1 - \alpha_t}}\epsilon_\theta(x_t, t)\right). \tag{10}$$

Once we have a trained $\epsilon_\theta(x_t, t)$, we can using the following sample method

$$x_{t-1} = \mu_\theta(x_t, t) + \sigma_t z, \quad z \sim \mathcal{N}(\mathbf{0}, \boldsymbol{I}). \tag{11}$$

In DDIM sampling [17], a denoising process could become deterministic when set $\sigma_t = 0$.

## B  Details of the Attention Pipeline

We present a comparative analysis of attention injection methods. As depicted in Fig. 15, MasaCtrl [23], while also adopting a self-attention injection approach, employs a more complex control mechanism in its second stage. In the first stage of MasaCtrl, the inverted latent representation $X_T^R$ is directly utilized by applying a modified prompt. In the second stage, a cross-attention mask is introduced to control specific word concepts modified in the prompt, which requires an additional forward pass. In contrast, our proposed method, RIVAL, primarily focuses on generating inconsistent variations. Consequently, we aim to guide feature interaction by replacing $KV$ features with an aligned latent distribution. Unlike MasaCtrl, our approach does not limit content transfer through editing prompts with only a few words. Hence, in the second stage, we employ a single forward pass without calculating an additional cross-attention mask, allowing fast and flexible text-to-image generation with diverse text prompts.

In recent updates, ControlNet [55] has incorporated an attention mechanism resembling the second stage of RIVAL to address image variation. However, a notable distinction lies in using vanilla noised latents as guidance, leading to a process akin to the attention-only approach employed in RePaint [24] with the Stable Diffusion model. Consequently, this methodology is limited to generating images within the fine-tuned training data domain.

## C  More About Comparisons

**Implementation Details.**  We compare our work with ELITE [20], Stable Diffusion image variation [53], and DALL·E 2 [16]. We utilize the official demo of ELITE to obtain results. To extract

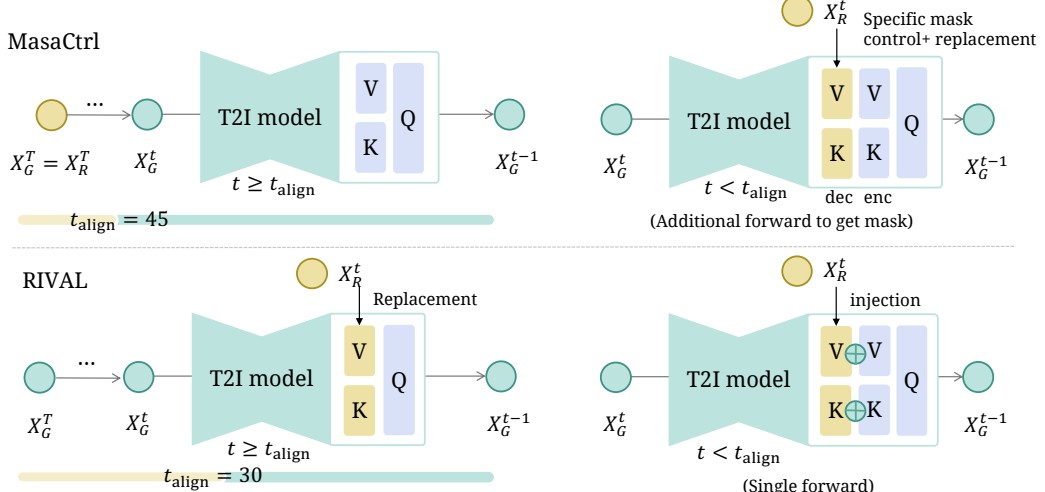

Figure 15: Self-attention control compare with MasaCtrl [23]. The default split of two stages is shown as a bar for each method.

| Experiments | AI | AF | LI | NA | Palette Distance | Text Alignment | Image Alignment |
|---|---|---|---|---|---|---|---|
| Baseline | | | | | 3.917 | 0.266 | 0.751 |
| AI | ✓ | | | | 3.564 | 0.277 | 0.804 |
| AF | ✓ | ✓ | | | 3.518 | 0.274 | 0.820 |
| LI. | | | ✓ | | 3.102 | 0.268 | 0.764 |
| NA | | | | ✓ | 3.661 | 0.251 | 0.647 |
| w/o AI&AF | | | ✓ | ✓ | 2.576 | 0.276 | 0.760 |
| w/o AF&NA | ✓ | | ✓ | | 2.419 | **0.279** | 0.817 |
| w/o AF | ✓ | | ✓ | ✓ | 1.902 | 0.274 | 0.818 |
| w/o LI | ✓ | ✓ | | ✓ | 3.741 | 0.242 | 0.653 |
| w/o NA | ✓ | ✓ | ✓ | | 2.335 | 0.267 | 0.839 |
| Full Model | ✓ | ✓ | ✓ | ✓ | **1.810** | 0.268 | **0.846** |

Table 3: Module-wise ablation (AI: Attention Injection, AF: Attention Fusion, LI: Latent Initialization, NA: Noise Alignment) with quantitative results.

context tokens, we mask the entire image and employ the phrase *"A photo/painting of ."* based on the production method of each test image. Inference for ELITE employs the default setting with denoising steps set to $T = 300$. For Stable Diffusion's image variation version, we utilize the default configuration, CFG guidance $m = 3$, and denoising steps $T = 50$. In the case of DALL·E 2, we utilize the official image variation API, specifically requesting using the most advanced API available to generate images of size $1024 \times 1024$.

**Comparison with UnCLIP.** UnCLIP [16], also known as DALL·E 2, is an image generation framework trained using image CLIP features as direct input. Thanks to its large-scale training and image-direct conditioning design, it generates variations solely based on image conditions when adapted to image variation. However, when faced with hybrid image-text conditions, image-only UnCLIP struggles to produce satisfactory results, particularly when CLIP does not recognize the image content correctly. We provide comparative analysis in Fig. 16. Additionally, we demonstrate in the last two columns of Figure 16 that our approach can enhance the accuracy of low-level details in open-source image variation methods such as SD image variation [53].

**Additional Visual Results.** We showcase additional results of our techniques in variation generation, as illustrated in Fig. 17, and text-driven image generation with image condition, as shown in Fig. 18. The results unequivocally demonstrate the efficacy of our approach in generating a wide range of image variations that accurately adhere to textual and visual guidance.

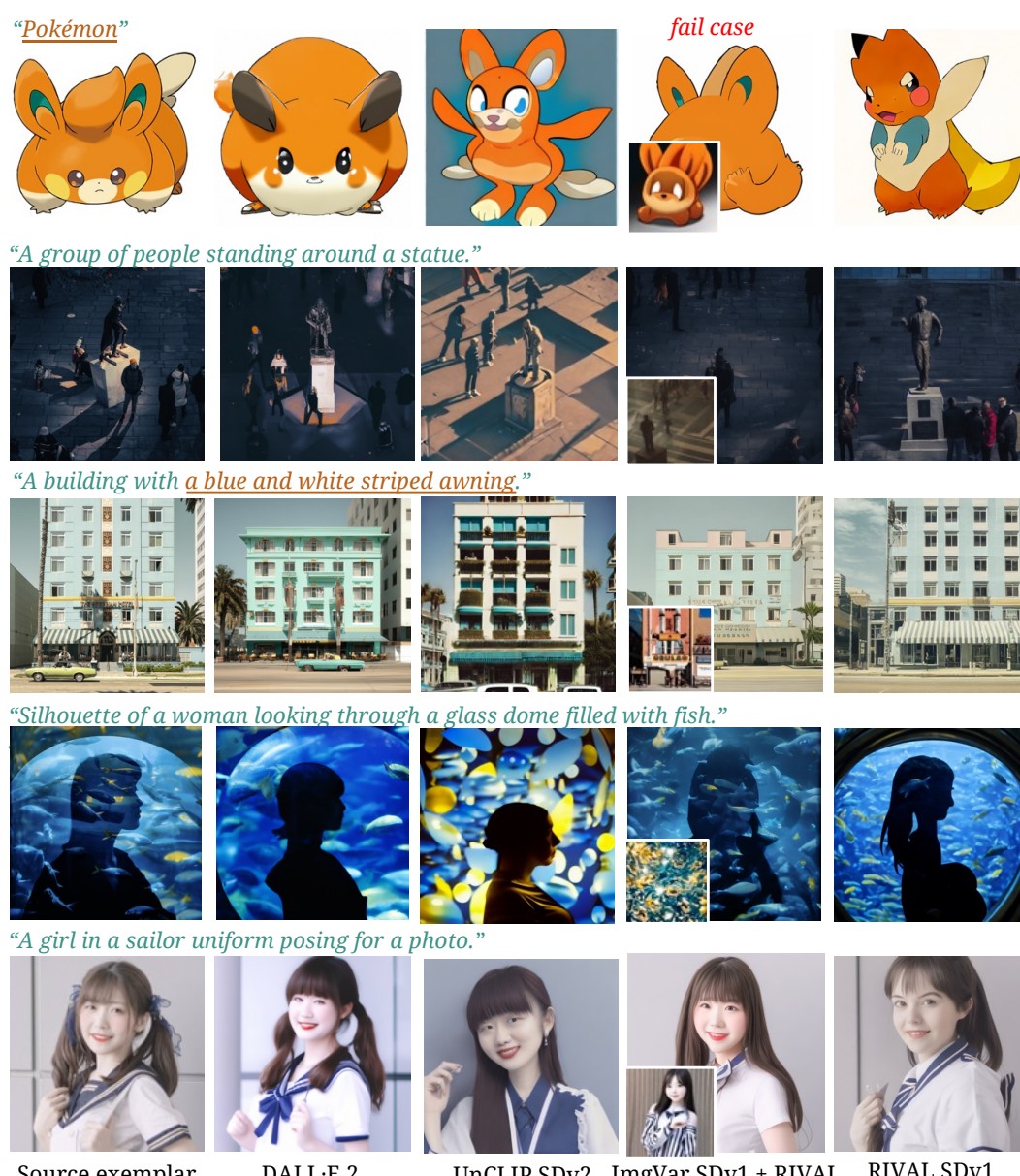

Figure 16: Comparision and adaptation with UnCLIP [16]. We highlight texts that enhance the image understanding for each case. Our inference pipeline is adapted to the image variation model depicted in the fourth column, in contrast to the variation achieved through vanilla inference in the bottom left corner of each image.

| $(t_{align}, t_{early})$ | (30, 30) | (0, 30) | (30, 0) | (0, 0) | (30, 50) | (50, 30) | (50, 50) |
|---|---|---|---|---|---|---|---|
| Text Alignment | 0.268 | 0.261 | 0.269 | 0.259 | 0.267 | 0.274 | **0.279** |
| Image Alignment | 0.846 | **0.873** | 0.838 | 0.865 | 0.839 | 0.813 | 0.817 |
| Palette Distance | 1.810 | **1.421** | 1.806 | 1.483 | 2.359 | 2.061 | 2.419 |

Table 4: Quantitative ablation for different alignment timesteps.

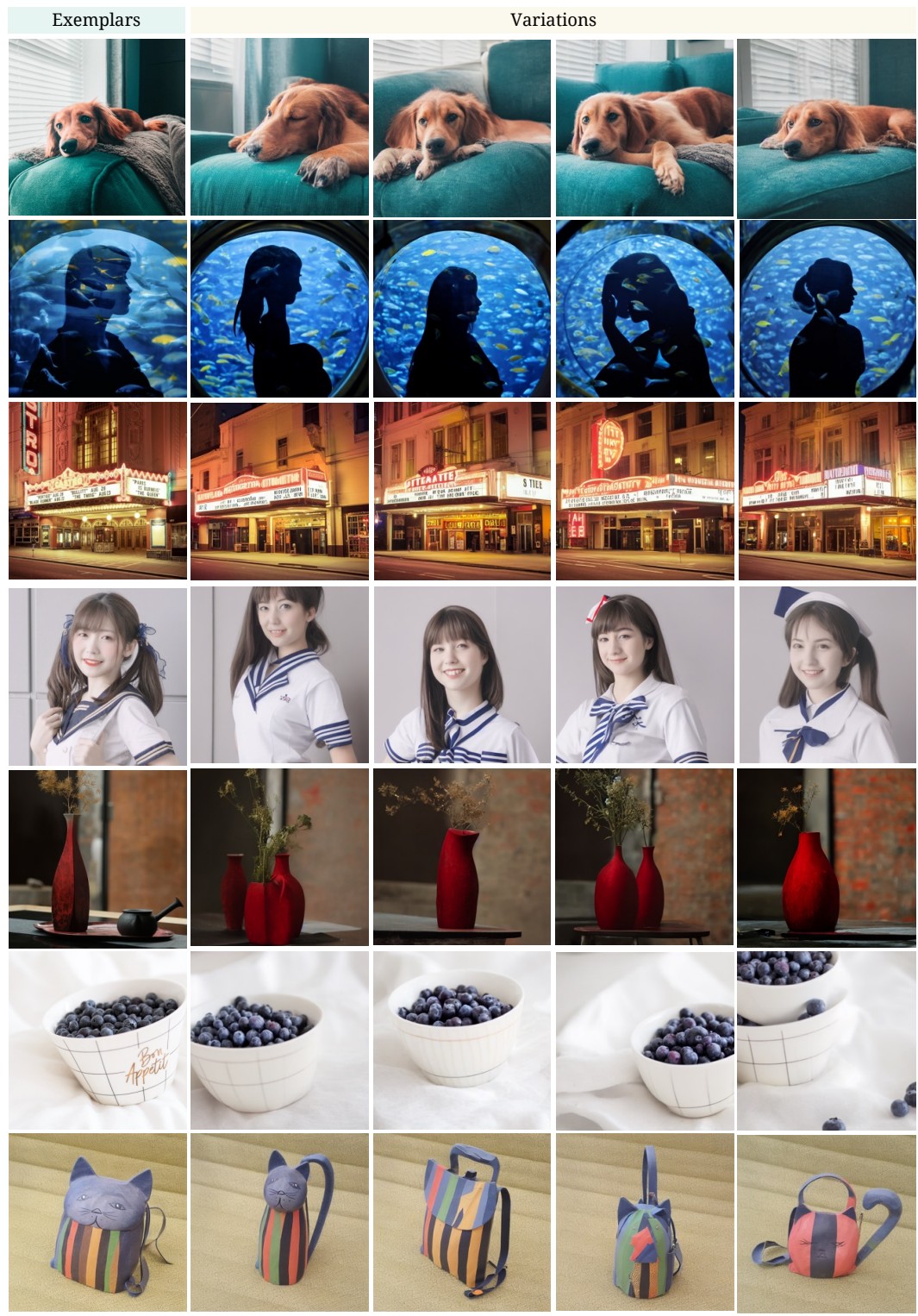

Figure 17: Text-driven free-form image generation results. The image reference is in the left column. In the last row, we also present variations for one customized concept `<sks> bag`.

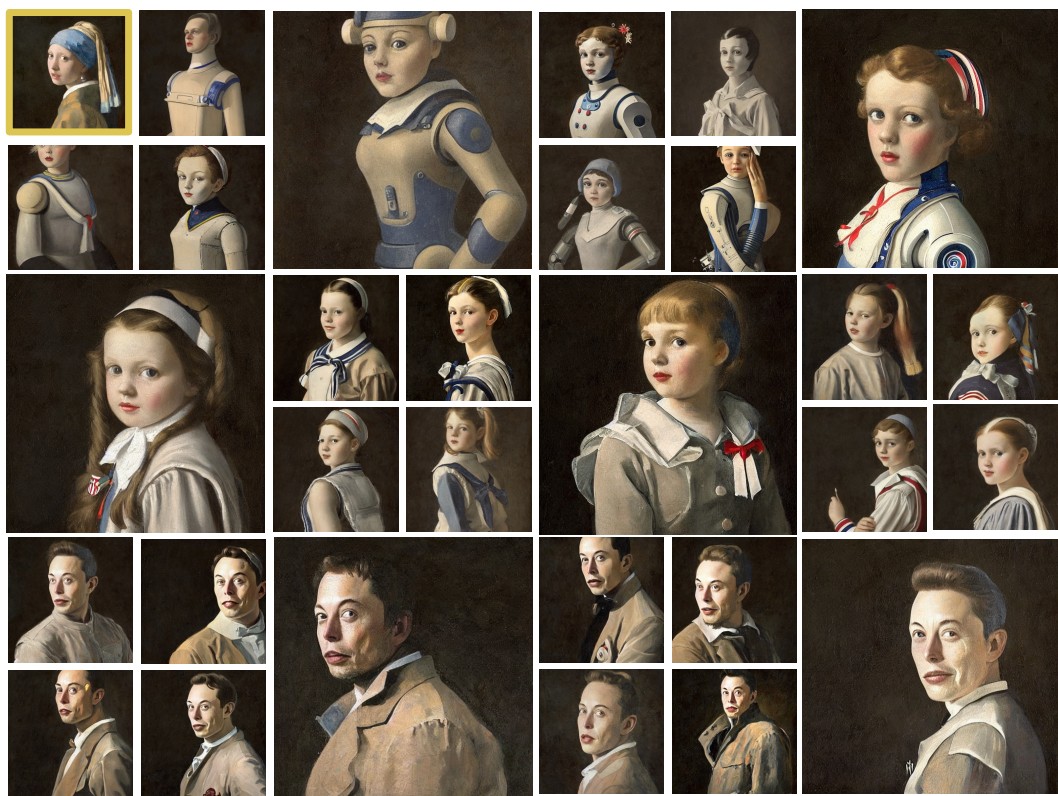

Figure 18: Text-driven free-form image generation results, with the image reference placed in the top left corner. The text prompts used are identical to those presented in Fig. 6 of the main paper. Every two rows correspond to a shared text prompt.

| CFG scale | 3 | 5 | 7 | 9 | 11 |
|---|---|---|---|---|---|
| Text Alignment | 0.260 | 0.272 | **0.273** | 0.273 | 0.271 |
| Image Alignment | 0.859 | **0.863** | 0.845 | 0.845 | 0.838 |
| Palette Distance | 1.749 | **1.685** | 1.737 | 1.829 | 1.902 |

Table 5: Quantitative ablation for different Classifier-Free Guidance Scale.

## D Additional Ablation Results

**Module-wise ablation.** We augment our visual ablations in Fig. 9 with a comprehensive module-wise experiment. The results in Tab. 3 underscore the role of each component and their combinations. Attention injection facilitates high-level feature interactions for better condition alignment (both text and image). Early fusion, built upon attention injection, aids in early step chain alignment, significantly enhancing image alignment. Meanwhile, latent noise alignment guarantees the preservation of color. Latent initialization has a pronounced impact, notably enhancing the color palette metric, an effect intensified by noise alignment.

**Ablation on early fusion step.** In addition to Fig. 10 of the main paper, we present comprehensive early-step evaluation results based on a grid search analysis in Fig. 19. By decreasing the duration of the feature replacement stage (larger $t_{\mathrm{align}}$), we observe an increase in the similarity of textures and contents in the generated images. However, excessively long or short early latent alignment durations ($t_{\mathrm{early}}$) can lead to color misalignment. Users can adjust the size of the early fusion steps as hyperparameters to achieve the desired outcomes.

We also conduct a quantitative evaluation in Tab. 4. It shows that there is a balance between two conditions (text-alignment and image-alignment). Besides, low-level texture-pasting artifacts will

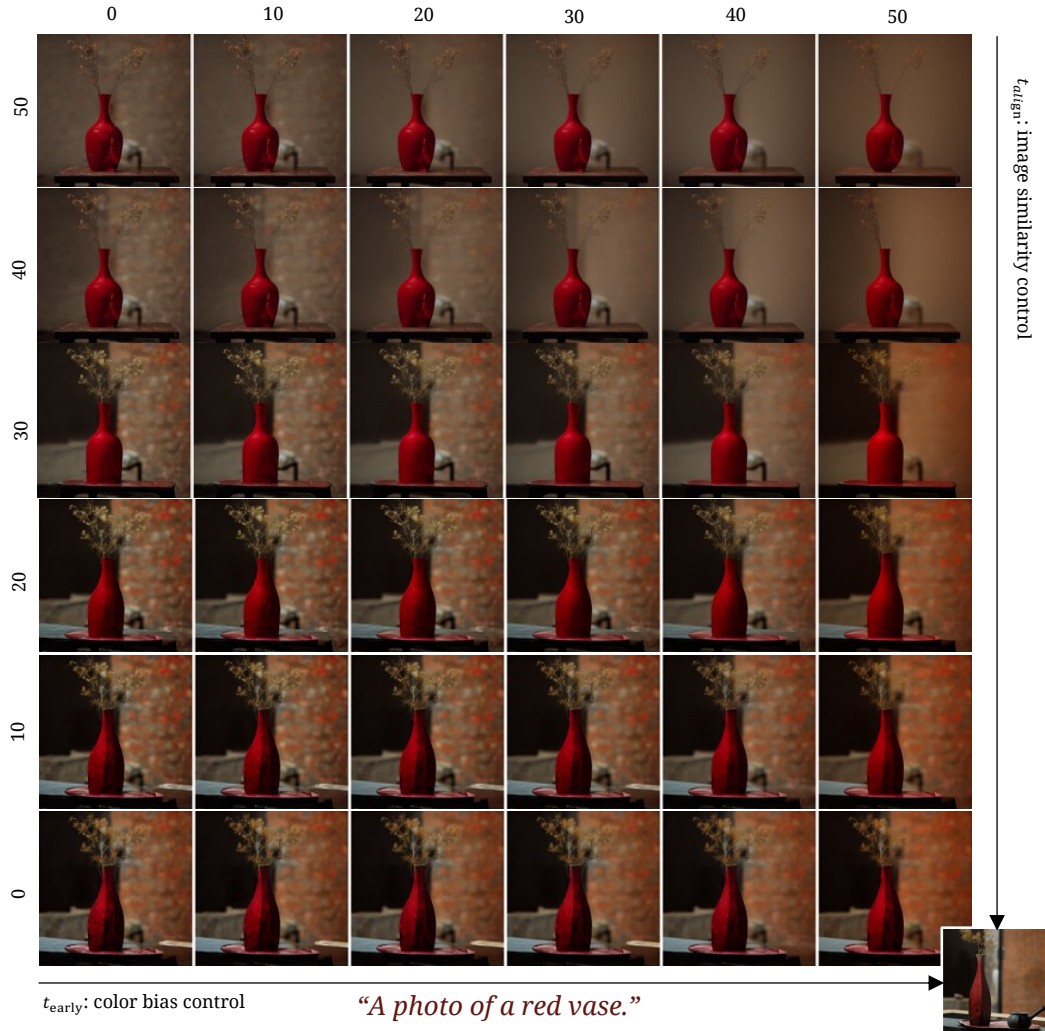

Figure 19: Ablation results for alignment steps, with the reference exemplar at the bottom right. We fix each generation's initial latent $X_G^T$.

present when $t_{align}$ is small. The choice of $t_{early}$ primarily influences the generated images' style (color) bias.

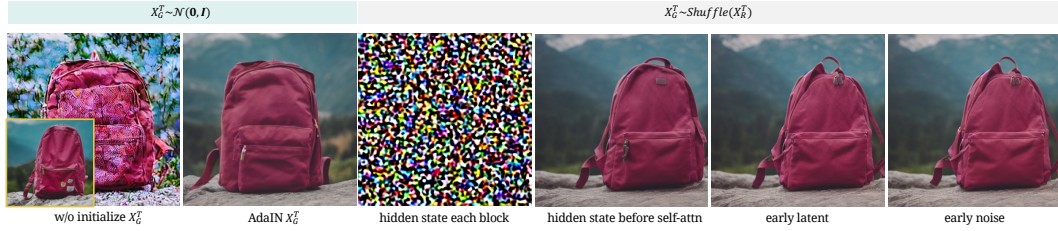

Figure 20: Ablation studies for different feature alignment strategies.

**Ablation on different alignment designs.**     Fig. 20 illustrates ablations conducted on various alignment designs. Two latent initialization methods, as formulated in Eq. (6), exhibit comparable performance. Nevertheless, incorporating alignments in additional areas, such as hidden states within each transformer block, may harm performance. Hence, we opt for our RIVAL pipeline's simplest noise alignment strategy.

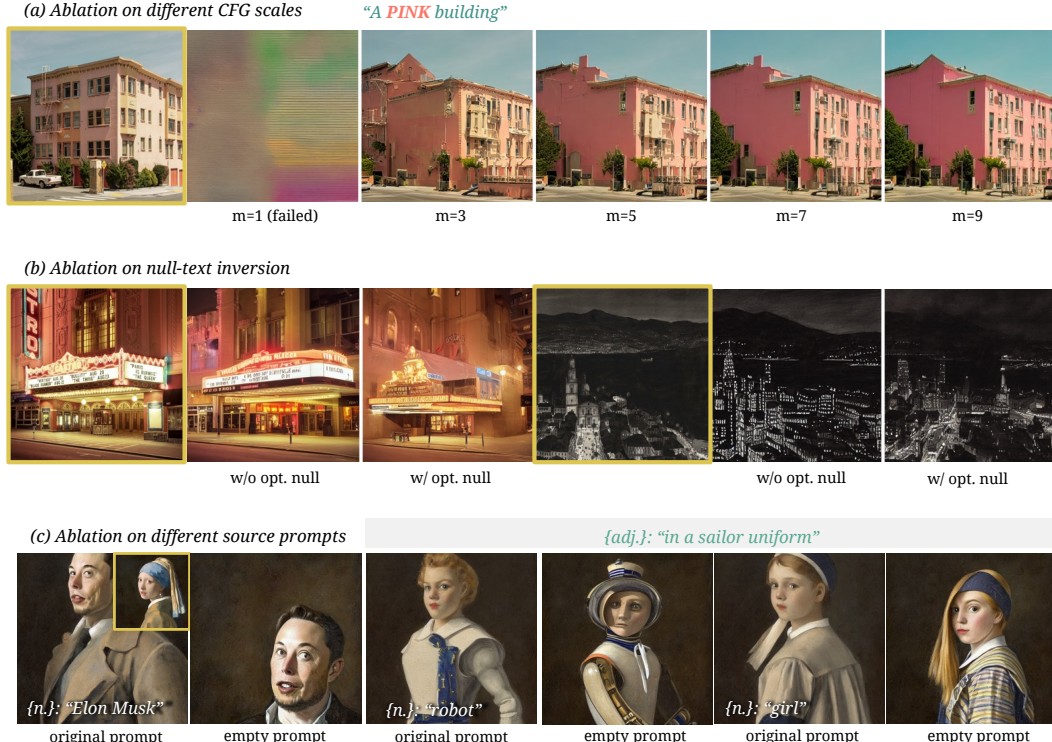

Figure 21: Ablation studies on different text conditions and guidance scales. Reference exemplars are highlighted with a golden border.

**Ablation on different text conditions.** We conduct ablations on text conditions in three aspects. First, we evaluate the impact of different CFG scales $m$ for text prompt guidance, with quantitative results in Tab. 5. As shown in Fig. 21 (a), our latent rescaling technique enables control over the text guidance level while preserving the reference exemplar's low-level features. Second, we employ an optimization-based null-text inversion method [18] to obtain an inversion chain with improved reconstruction quality. However, this method is computationally intensive, and the optimized embeddings are sensitive to the guidance scale. Furthermore, when incorporating this optimized embedding into the unconditional inference branch, there is a variation in generation quality, as depicted in Fig. 21 (b). Third, we utilize empty text as the source prompt to obtain the latents in the inversion chain while keeping the target prompt unchanged. As depicted in Fig. 21 (c), the empty text leads to weak semantic content correspondence between the two chains but sometimes benefits text-driven generation. For example, if users do not want to transfer the "gender" concept to the generated robot.

## E  Quantitative Evaluations

This section comprehensively evaluates our proposed method with various carefully designed metrics, including CLIP Score, color palette matching, user study, and KID.

**CLIP Score.** For evaluating the CLIP Score, we employ the official ViT-Large-Patch14 CLIP model [56] and compute the cosine similarity between the projected features, yielding the output.

**Color palette matching.** To perform low-level matching, we utilize the Pylette tool [59] to extract a set of 10 palette colors. Subsequently, we conduct a bipartite matching between the color palette of each generated image and the reference palette colors in the RGB color space. Before matching, each color is scaled to $[0, 1]$. The matching result is obtained by calculating the sum of L1 distances.

**User study.** To evaluate the effectiveness of our approach against other methods, we conducted a user study using an online form. The user study interface, depicted in Figure 22, was designed to elicit

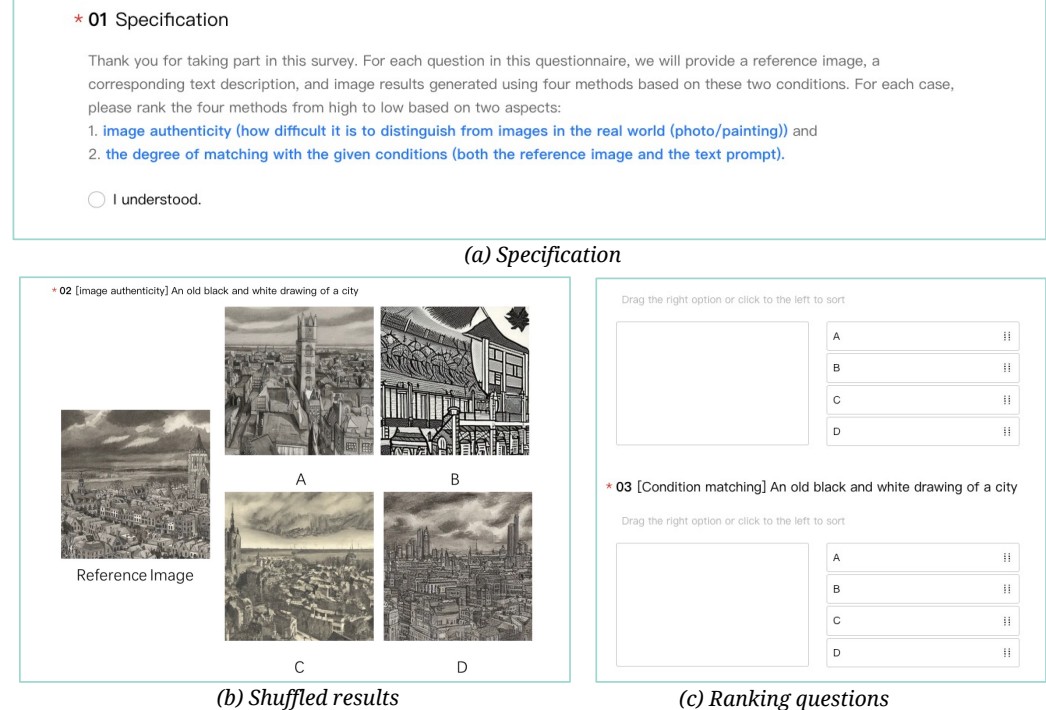

*(a) Specification*

*(b) Shuffled results*     *(c) Ranking questions*

Figure 22: User study user interface. In this case, four methods are: (A). SD ImageVar [53], (B). ELITE [20], (C). DALL·E 2[16], (D). RIVAL (ours).

| method | SD [11] | ImgVar [53] | ELITE [20] | UnCLIP [16] | **RIVAL** |
|---|---|---|---|---|---|
| base model | V1-5 | V1-3 | V1-4 | V2-1 | V1-5 |
| KID ↓ | 17.1 | 18.5 | 25.7 | 13.5 | **13.2** |

Table 6: Quantitative comparisons for KID ($\times 10^3$). All methods are Stable Diffusion based.

user rankings of image variation results. We collected 41 questionnaire responses, encompassing 16 cases of ranking comparisons.

**KID evaluation.** To provide a comprehensive assessment of the quality, we utilize Kernel Inception Distance (KID)[60] to evaluate the perceptual generation quality of our test set. As depicted in Table6, with Stable Diffusion V1-5, our method achieves the best KID score, which is slightly superior to the UnCLIP [16], employing the advanced Stable Diffusion V2-1.

# F    Additional Considerations

**Data acquisition.** To comprehensively evaluate our method, we collected diverse source exemplars from multiple public datasets, such as DreamBooth [12] and Interactive Video Stylization [61]. Some exemplars were obtained from Google and Behance solely for research purposes. We will not release our self-collected example data due to license restrictions.

**Societal impacts.** This paper introduces a novel framework for image generation that leverages a hybrid image-text condition, facilitating the generation of diverse image variations. Although this application has the potential to be misused by malicious actors for disinformation purposes, significant advancements have been achieved in detecting malicious generation. Consequently, we anticipate that our work will contribute to this domain. In forthcoming iterations of our method, we intend to introduce the NSFW (Not Safe for Work) test for detecting possible malicious generations. Through rigorous experimentation and analysis, our objective is to enhance comprehension of image generation techniques and alleviate their potential misuse.

