*"Silhouette of a woman looking through a glass dome filled with fish."*

*"A girl in a sailor uniform posing for a photo."*

Source exemplar   DALL·E 2   UnCLIP SDv2 ImgVar SDv1 + RIVAL RIVAL SDv1

Figure 12: Comparision and adaptation with UnCLIP [8]. We highlight texts that enhance the image understanding for each case. Our inference pipeline is adapted to the image variation model depicted in the fourth column, in contrast to the variation achieved through vanilla inference in the bottom left corner of each image.

null-text inversion method [5] to obtain an inversion chain with improved reconstruction quality. However, this method is computationally intensive, and the optimized embeddings are sensitive to the guidance scale. Furthermore, when incorporating this optimized embedding into the unconditional inference branch, there is a variation in generation quality, as depicted in Fig. 17 (b). Third, we utilize empty text as the source prompt to obtain the latents in the inversion chain while keeping the target prompt unchanged. As depicted in Fig. 17 (c), the empty text leads to weak semantic content correspondence between the two chains but sometimes benefits text-driven generation. For example, if users do not want to transfer the "gender" concept to the generated robot.

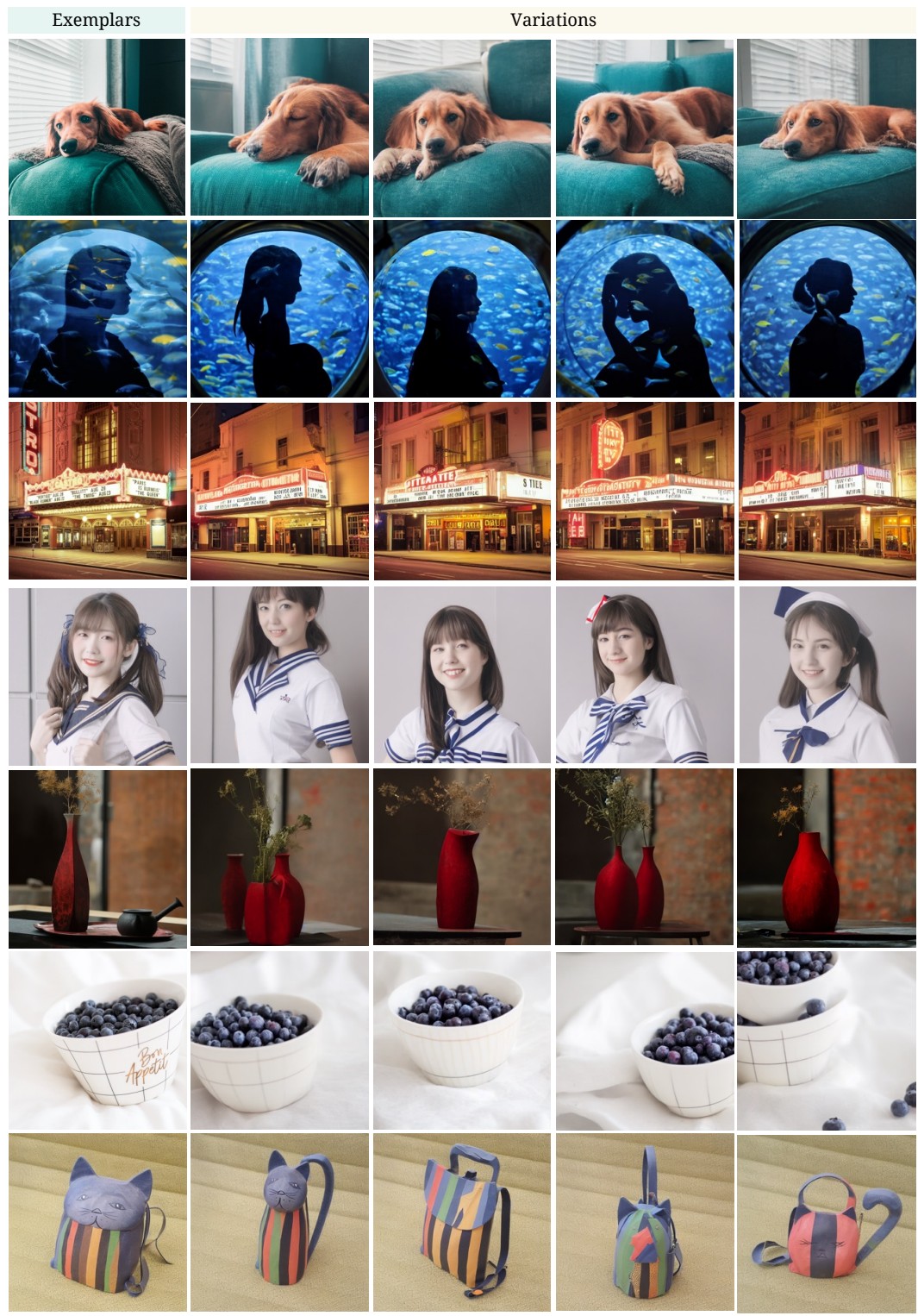

Figure 13: Text-driven free-form image generation results. The image reference is in the left column. In the last row, we also present variations for one customized concept `<sks> bag`.

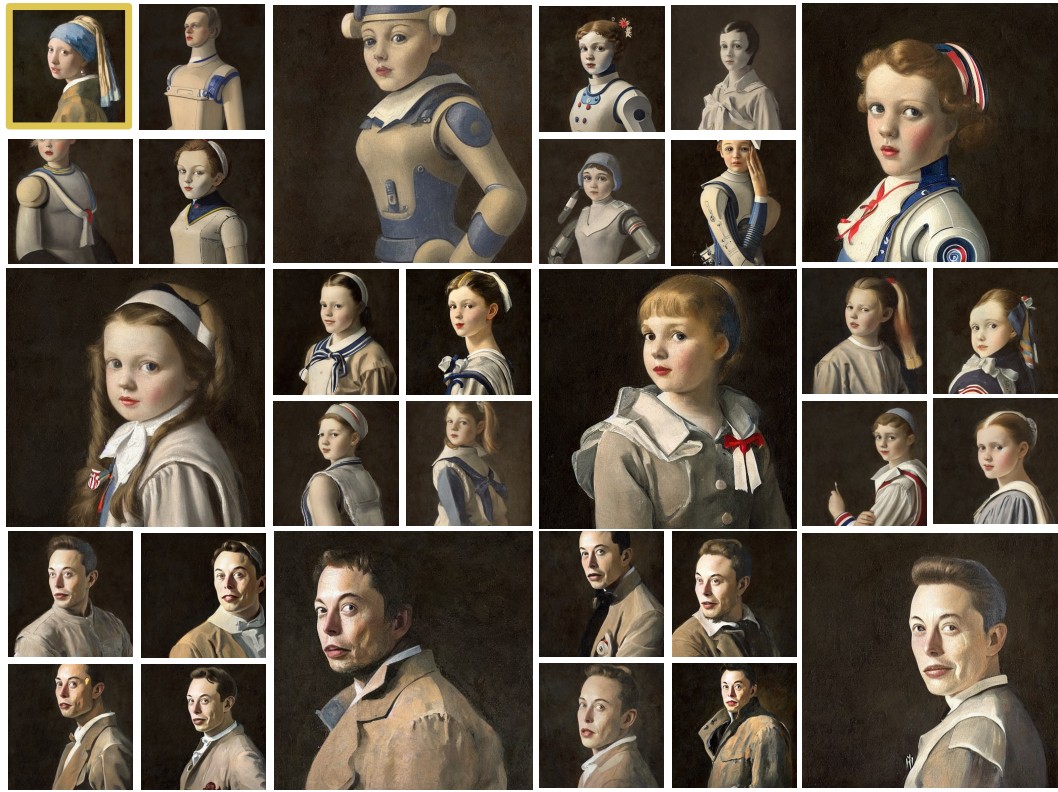

Figure 14: Text-driven free-form image generation results, with the image reference placed in the top left corner. The text prompts used are identical to those presented in Fig. 5 of the main paper. Every two rows correspond to a shared text prompt.

| method | SD [9] | ImgVar [6] | ELITE [14] | UnClIP [8] | **RIVAL** |
|---|---|---|---|---|---|
| base model | V1-5 | V1-3 | V1-4 | V2-1 | V1-5 |
| KID $\downarrow$ | 17.1 | 18.5 | 25.7 | 13.5 | **13.2** |

Table 2: Quantitative comparisons for KID ($\times 10^3$). All methods are Stable Diffusion based.

## E  Quantitative Evaluations

This section comprehensively evaluates our proposed method with various carefully designed metrics, including CLIP Score, color palette matching, user study, and KID.

**CLIP Score.** For evaluating the CLIP Score, we employ the official ViT-Large-Patch14 CLIP model [7] and compute the cosine similarity between the projected features, yielding the output.

**Color palette matching.** To perform low-level matching, we utilize the Pylette tool [12] to extract a set of 10 palette colors. Subsequently, we conduct a bipartite matching between the color palette of each generated image and the reference palette colors in the RGB color space. Before matching, each color is scaled to $[0, 1]$. The matching result is obtained by calculating the sum of L1 distances.

**User study.** To evaluate the effectiveness of our approach against other methods, we conducted a user study using an online form. The user study interface, depicted in Figure 18, was designed to elicit user rankings of image variation results. We collected 41 questionnaire responses, encompassing 16 cases of ranking comparisons.

**KID evaluation.** To provide a comprehensive assessment of the quality, we utilize Kernel Inception Distance (KID)[1] to evaluate the perceptual generation quality of our test set. As depicted in Table 2, with Stable Diffusion V1-5, our method achieves the best KID score, which is slightly superior to the UnCLIP [8], employing the advanced Stable Diffusion V2-1.

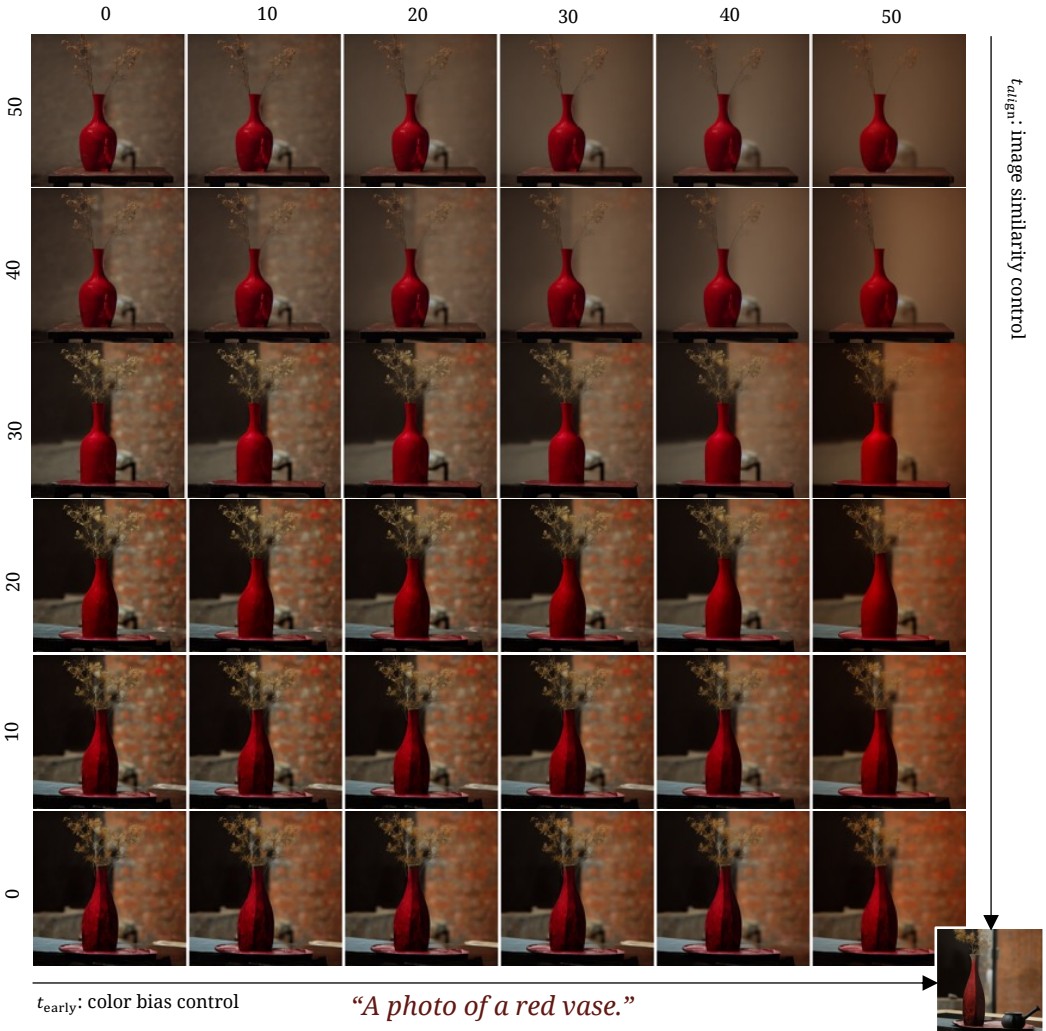

Figure 15: Ablation results for alignment steps, with the reference exemplar at the bottom right. We fix each generation's initial latent $X_G^T$.

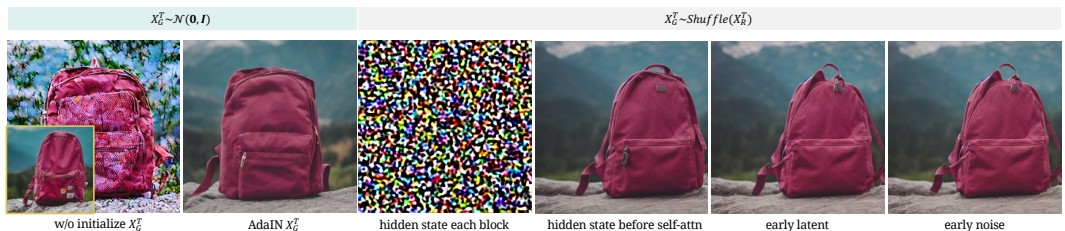

Figure 16: Ablation studies for different feature alignment strategies.

## F  Additional Considerations

**Correction of an equation error in the main paper.** In the main paper, it has been identified that an error exists in Equation (4). The residual should be applied after completing the entire self-attention process. Therefore, the updated output of the hidden state in the self-attention mechanism is expressed as follows:

$$\mathbf{v}_G^* = \text{softmax}\left(\frac{QK^\top}{\sqrt{d_k}}\right)V. \tag{14}$$

We will correct this equation in the updated version of the main paper.

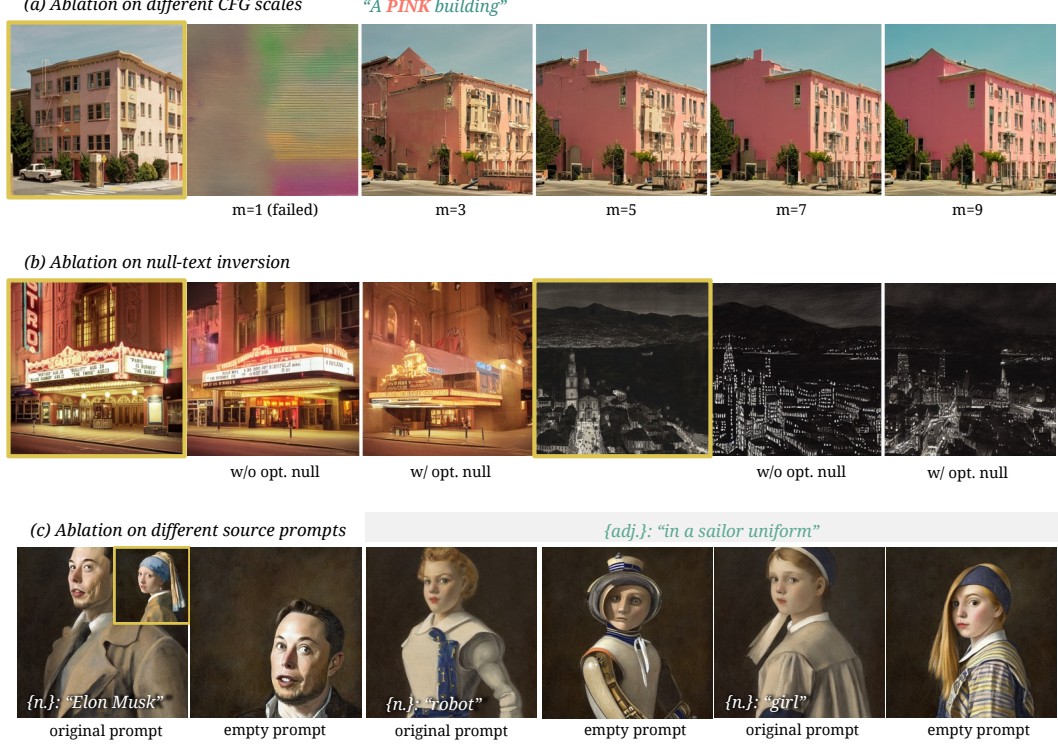

Figure 17: Ablation studies on different text conditions and guidance scales. Reference exemplars are highlighted with a golden border.

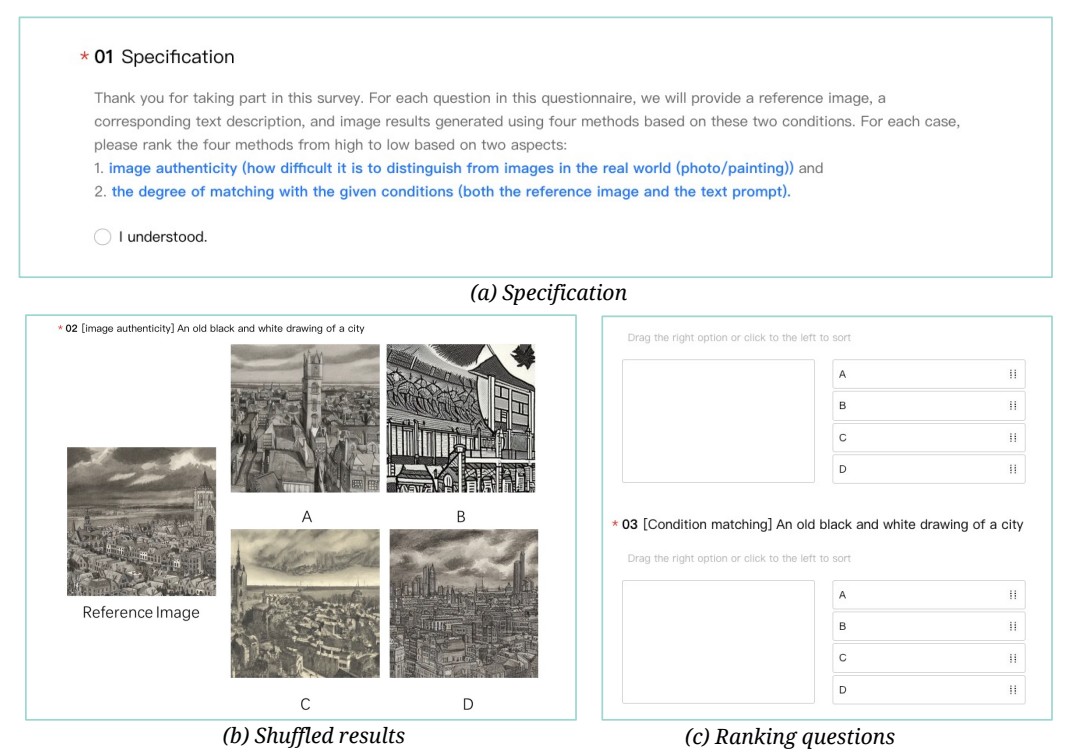

Figure 18: User study user interface. In this case, four methods are: (A). SD ImageVar [6], (B). ELITE [14], (C). DALL·E 2[8], (D). RIVAL (ours).

**Data acquisition.** To comprehensively evaluate our method, we collected diverse source exemplars from multiple public datasets, such as DreamBooth [10] and Interactive Video Stylization [13]. Some exemplars were obtained from Google and Behance solely for research purposes. We will not release our self-collected example data due to license restrictions.

**Societal impacts.** This paper introduces a novel framework for image generation that leverages a hybrid image-text condition, facilitating the generation of diverse image variations. Although this application has the potential to be misused by malicious actors for disinformation purposes, significant advancements have been achieved in detecting malicious generation. Consequently, we anticipate that our work will contribute to this domain. In forthcoming iterations of our method, we intend to introduce the NSFW (Not Safe for Work) test for detecting possible malicious generations. Through rigorous experimentation and analysis, our objective is to enhance comprehension of image generation techniques and alleviate their potential misuse.