# OpenReview forum: "Real-World Image Variation by Aligning Diffusion Inversion Chain"
_NeurIPS.cc/2023/Conference — NeurIPS 2023 spotlight_

### Official Review · Reviewer_5k69 · 2023-06-23

**Soundness:** 3 good
**Presentation:** 4 excellent
**Contribution:** 3 good
**Rating:** 7
**Confidence:** 3

**Summary:**

This paper proposes a framework RIVAL to generate variation of the real image without any tuning. It has two key components, the first one is a cross-image self-attention injection, where it first inverts the real image using DDIM, and then sample another random chain but uses a mix of real inverted chain’s value and denoised value in the denoising process. To mitigate the issue of out-of-distribution issue of the inverted latent of the real image, it further proposes a latent chain alignment process, where the random chain is initialised with the adaptively normalized Gaussian distribution of the inverted real latent. Experiments show improvement over baseline methods in terms of Text alignment and user preference. Overall the paper is well-written, the method is easy to follow.


**Strengths:**

The proposed method does not require training or finetuning, although the inference time becomes twice as long as the normal inference, it is acceptable in practice.

RIVAL is shown to work on other text-to-image applications such as test-driven image generation with real-image condition and example-based inpainting.

Qualitative results show better identity preservation and style matching to the reference image compared to other baselines, plus, the editing results on text conditioning is impressive, considering no finetuning is needed.

The authors also provide detailed ablation studies on the choice of alignment steps and framework components.


**Weaknesses:**

The author does not provide analysis on the edibility of the proposed method, which I think is an important feature.

How sensitive is the method to the CFG guidance weight? There is no ablation study on the choice of the CFG weight


**Questions:**

In Figure 4, the person's identity seems to change over regeneration, any idea on how to better preserve the identity?

Can the proposed method work with image conditional DM? Would be great to see some visual examples


**Limitations:**

The authors provide limitation analysis, and see my questions above for further clarification

---

> ### Author Rebuttal · Authors · 2023-08-09
>
> Dear reviewer 5k69,
>
> Thank you for your valuable feedback and constructive comments. We will address your comments below and in the revised paper.
> ### Q1 A Clarification of Inference Time
> We want to clarify the point on inference time:
>
> After the inversion process, the overall inference time is not doubled. This claim is attributed to the concurrent execution of the denoising steps for both the inversion and generation chains when processing batched inputs, as illustrated in Fig. 2. The primary computational overhead emerges during the self-attention phase for $t>t_{\text{align}}$, due to a doubling in QKV matrix multiplication from the increased size of KV (Eq. 4).
>
> As mentioned in **aMYr@Q3**, in our experiment, we've cut RIVAL's inference time down to 6 seconds by employing half-precision and x-formers, maintaining 50 DDIM steps.
>
> Furthermore, efficiency could be further improved by caching hidden states of the inversion chain in the inversion process.
>
> ### Q2 Sensitivity of the CFG Scale
> CFG scale in our diffusion model is an essential hyperparameter controlling the intensity of guidance. We have conducted a visual ablation study to examine the effect of various CFG scales, and the results are presented in Appendix Fig. 17 (a). The study reveals that the CFG scale influences the variations generated by our model, for instance, the color of the building in Appendix Fig. 17 (a). We performed a quantitative ablation on our dataset with different CFG values. It's worth noting that even with large CFG scales, our generated images did not exhibit color artifacts, as demonstrated in **rebuttal Fig. H**.
> |CFG scale|3|5|7|9|11|
> |-|-|-|-|-|-|
> |Text Alignment  |0.260|0.272|**0.273**|0.273|0.271|
> |Image Alignment |0.859|**0.863**|0.845|0.845|0.838|
> |Palette Distance|1.749|**1.685**|1.737|1.829|1.902|
>
> As shown in the above table, our RIVAL maintains a balance between the strength of the text guidance and the preservation of features from the reference exemplar, thanks to the proposed latent noise rescaling mechanism in Eq. (9-10).
> ### Q3 Identity Preservation in Fig. 4
> Identity preservation, especially in the context of tuning-free customization methods such as our RIVAL, ELITE, and BLIP Diffusion, is indeed a challenging problem. Unlike customizing some "easy" concepts, like specific animal breeds and colors, preserving certain customizable concepts, like personal identity or particular objects, requires intricate control and optimization.
>
> One feasible solution could be to integrate RIVAL with customized models. We provide an illustrative example in Fig. 9 and the last row of Appendix Fig. 13, where we combined RIVAL with DreamBooth's [sks] doll model. This ability demonstrates how our approach can work synergistically with customized models to achieve better identity preservation, and we will continually explore the direction of tuning-free personalization in our future work.
>
> ### Q4 Compatibility with Image Conditional Diffusion Models
> Regarding image conditional diffusion models, we recognize two primary branches. The first is represented by models like UnCLIP (DALLE-2), which accepts an image as a semantic condition. As an example of this type of model's integration with RIVAL, we point to Appendix Fig. 12 (4th column), where we use the open-sourced ImgVar SDv1[29]. RIVAL is able to maintain the tone and low-level style consistency of the reference image, which contrasts with images generated solely using ImgVar (shown in each left-bottom corner of each image).
>
> The second branch of image conditional diffusion models consists of controllable diffusion models like ControlNet. In this context, we demonstrate the application of the RIVAL integration with ControlNet and present examples in **rebuttal Fig. B**.
> The implementation is straightforward. We adapt RIVAL with ControlNet by adding condition residuals of ControlNet blocks on the generation chain. Results illustrate that RIVAL, as a plug-and-play module, exhibits strong generalization capabilities when combined with various image-conditioned models.
>
> ### Q5 "Editability" of RIVAL
> Concerning "editability", we understand it in three different ways. If your interpretation is different from ours, we would appreciate your feedback.
>
> 1. **Model adaptability**: This refers to the editability of our method to adapt to other models. As discussed in Q4, our approach is highly adaptable across several popular frameworks, such as UnCLIP, ControlNet, and the recent SDXL. The results are shown in **rebuttal Fig. A&B**.
> 2. **Text-conditioned image editing**: This refers to the editability for the text-conditioning image editing and generation task. We provide non-cherry-picked visual examples in Appendix Fig. 14 and **rebuttal Fig. C&E**. We also present some ablations concerning editability in Appendix Fig. 17 (c), focusing on selecting source prompts. Using an empty prompt for inversion, we achieve results with more flexibility and consistency with the text prompt. Among those experiments, RIVAL produces results that consistently adhere to the reference image's low-level attributes while maintaining a high-quality match with the text prompt.
> 3. **Adjustability of the method itself**: This pertains to the editability of the method in terms of controlling the generation process. Our method provides several editable hyperparameters. For instance, the alignment steps $t_{align}$ and $t_{early}$ can be adjusted (a visual ablation is shown in Appendix Fig. 15, **quantitative results in aMYr@Q5**). Additionally, the target prompt (in image editing) and the CFG value can be adjusted for better text alignment, as mentioned in Q2.

---

> ### Comment · Reviewer_5k69 · 2023-08-21
> **Thank you for your response**
>
> The response resolved my concerns, I retain my original rating. I encourage the author to release code for the community.

---

> > ### Author Response · Authors · 2023-08-21
> > **Thank You for Recognizing RIVAL**
> >
> > Dear Reviewer 5k69,
> >
> > We are glad to see our rebuttal well addressed your issues. We greatly appreciate your consistently positive feedback on our work from beginning to end. Your comments are meaningful; for instance, considerations like identity preservation and the impact of CFG have led us to valuable reflections.
> >
> > As mentioned in our abstract, we will update the code after the final decisions are confirmed. We will release the code link and corresponding implementation in the final version (including new applications mentioned in the rebuttal, such as ControlNet).
> >
> > Thank you again for recognizing RIVAL!

---

### Official Review · Reviewer_SLMY · 2023-07-03

**Soundness:** 3 good
**Presentation:** 2 fair
**Contribution:** 3 good
**Rating:** 5
**Confidence:** 4

**Summary:**

This paper works on the design of diffusion model to generate image variations given an image examplar as the source image.  The basic idea is to align the image generation process to the image inversion process of source image. This is achieved by designing an cross-image self-attention injection for feature interaction between the source and generated images in the diffusion process. The results demonstrate better performance compared with the existing methods in image-conditioned text-to-image generation and examplar-based image inpainting.

**Strengths:**

The idea to generate image variations given a source image is an interesting task. This paper proposed to align the source image inversion process and image generation process using the cross-image self-attention injection, which is a simple idea and shows good image generation results.

**Weaknesses:**

1. The presentation and training of the parameters in "cross-image self-attention injection" are unclear to me. I understand that, in sect. 3.1, the features of examplar image and the generated image are interacted based on eqns. (2-4), in their inversion and generation chain respectively. However, how to train/determine the parameters in these equations? Are these parameters shared across different diffusion steps?  And how do these parameters affect the final generation results?

2. Does this proposed generation method rely on the off-the-shelf diffusion model without any training? The chain alignment is conducted in the latent feature space or the image space?

3. An algorithmic description of this proposed method for image generation should be given for better understanding the training (possibly) and image generation process?

4. The shuffle operation in sect. 3.2 is described as "permutated sample from the inverted reference latent". Please clarify on this description, does this mean permutating the pixels of X_R^T?

5. In the Table I, please clarify on the metric of "Real-image Rank" and "Preference Rank".  I know that lines 228-233 may describe them, however, it is still unclear to me on the difference between "Real-image Rank" and "Preference Rank".

**Questions:**

I am interested in the problem to tackle and the proposed idea of feature interaction between source and generated image for inverse and generation chain alignment. However, I have many unclear points on the presentation of the proposed method (as mentioned in the weakness). I suggest authors to clarify the method in rebuttal and improve the writings and presentations of this manuscript.

**Limitations:**

The paper discussed on the limitation of this work.

---

> ### Author Rebuttal · Authors · 2023-08-09
>
> Dear reviewer SLMY,
>
> Thank you for your valuable feedback and constructive comments. We will address your comments below and in the revised paper.
> ### Q1 Clarification of Cross-Image Self-Attention Injection
> We employ the pre-trained Stable Diffusion model in RIVAL without adding trainable parameters. All parameters are consistent across steps. Our self-attention injection design modifies the input and interactions of self-attention. Here we consider each chain in the forward process separately:
>
> 1. **Inversion Chain**: Because we want to ensure that this chain reconstructs the inverted image, the denoising process is identical to the vanilla diffusion generation forward. No modifications are made in this chain. We use the hidden state feature $\mathrm{v}_R$ before each self-attention for attention injection to the Generation Chain.
>
> 2. **Generation Chain**: For the early generation steps $t > t_\text{align}$, we replace the KV values with $W^V (\mathrm{v}_R) $, $W^K (\mathrm{v}_R) $ using the hidden state $\mathrm{v}_R$ from the inversion chain for self-attention calculation.  In the later generation steps, we concatenate $\mathrm{v}_R$ and the hidden state from the generation chain itself $\mathrm{v}_G$ to obtain new KV values. In all self-attention calculations, we do not change Q values and maintain them as $W^Q(\mathrm{v}_G)$. It's worth noting that all parameters $W^{(\cdot)}$ are frozen.
> ### Q2 Clarification of Generation Method
> #### Q2-1 Does this proposed generation method rely on the off-the-shelf diffusion model without training?
> As elaborated in response Q1, our training-free approach is built on the pre-trained Stable Diffusion model. Given its prowess in reconstructing real-world images using DDIM inversion, we infer its potential to generate images within a wide real-world domain. This foundation ensures RIVAL's plug-and-play functionality for arbitrary input images. Furthermore, we've showcased RIVAL's synergy with the recent diffusion model, SDXL, in **rebuttal Fig. A**.
> #### Q2-2 Is the chain alignment conducted in the latent feature or image space?
> In our approach, we align two chains in the latent feature space. A visual ablation study was conducted and presented in Appendix Fig. 16. One of the reasons is that the chains should be aligned at the initial state $X_G^T$ and $X_R^T$. In the image generation process, we found that aligning in the latent space has a similar effect as aligning the residual part in the self-attention. This result paves the way for RIVAL's adaptability to potential latent-free diffusion models. We align in the latent space for simplicity and computational efficiency by adjusting the predicted noise in both branches, as detailed in Eq. 9-10.
> ### Q3 Algorithmic Description
> Below, we provide a Python-style pseudocode for RIVAL. A LaTeX algorithm version will be included in the revision.
> ```py
> def RIVAL(ref_img, c, cfg, DM, T):
>     '''
>     ref_img: the reference image
>     c: text condition
>     cfg: classifier-free guidance scale
>     DM: a pre-trained diffusion model
>     T: number of total denoising steps
>     return -> the generated image
>     '''
>     t_align = t_early = int(T*0.6)
>     # Eq.(5-6), generate an inversion latent chain of T steps using DDIM
>     inv_chain = DDIM_inv(ref_img, c, T, DM)
>     x_rt = inv_chain[-1] # the last latent X_R^T
>     # Eq.(8)
>     x_gt = shuffle(x_rt) # initialized latent in the generation chain
>
>     # start denoising process, t from T to 0
>     for t in range(T, 0, -1):
>         # Inversion chain is the normal denoising process of the diffusion model
>         # pred_r and pred_g are noise term predictions for each step
>         pred_r, v_rs = DM.unet(x_rt, c, T, cfg=1)
>         pred_g = x_gt
>
>         # use blks to denote the unet blocks
>         for blks in DM.unet.blks:
>             v_g = pred_g
>             # Eq.(3)
>             if t > t_align:
>                 # v_r: residual hidden state of generation chain
>                 v_r = blk(v_g)
>                 v_kv = v_rs[blk]
>             else:
>                 v_r = blk(v_g) # [bs, ndim, HW]
>                 v_kv = torch.cat([v_rs[blk], v_r]) # [bs, ndim, 2*HW]
>
>             # Eq.(4)
>             pred_g = v_g + blk.self_attn(v_r, v_kv)
>             # Other operations in the unet are omitted for simplification
>             pred_g = blk.others(pred_g, c, T)
>
>         # Eq.(9), cfg rescale
>         pred_g = rescale(pred_g, cfg)
>
>         # Eq.(10), chain latent alignment
>         if t > t_early:
>             pred_g = AdaIN(pred_g, pred_r)
>
>         x_gt =  DDIM_scheduler(x_gt, pred_g, T)
>         x_rt =  DDIM_scheduler(x_rt, pred_r, T)
>
>     return DM.vae.decode(x_gt) # decode X_G^0 to get the variance results
> ```
> ### Q4 Explanation of the Shuffle Operation
> You are correct. The shuffle operation $X_G^T = \text{shuffle}(X_R^T)$ in Eq.8 is designed to perform a permutation in the pixel dimension. We'll ensure this is more explicitly stated in our revised version. Refer to the PyTorch-style code snippet in **4HxC@Q1**.
> ### Q5 Clarification of "Real-Image Rank" and "Preference Rank" Metrics
> The **Real-Image Rank** originates from the user study, where participants were asked to rank images according to their perceived authenticity - which image they believed was most likely not AI-generated.
>
> The **Preference Rank**, on the other hand, originates from the user study where participants, given a reference image and text prompt, ranked generated images based on their adherence to the image reference and the text description.
>
> We acknowledge the potential confusion between these terms and have renamed them "Real-World Authenticity Preference" and "Condition Adherence Preference" in the revision.
> ### Q6 Improving Presentation
> We appreciate your comments and advice on the presentation of the paper. We will include more details and refine our presentation to make the content more easily understandable in the revision.

---

> > ### Comment · Reviewer_SLMY · 2023-08-17
> > **Thanks for the responses**
> >
> > I appreciate the responses for clarifying my questions and they are mostly clear, and the paper should consider these clarifications in the revised version, especially on the Q1 and Q2.  I retain my rating and it is ok to me if accepting the paper.

---

> > > ### Author Response · Authors · 2023-08-17
> > > **Thank You for Your Comments and Feedback**
> > >
> > > Dear Reviewer SLMY,
> > >
> > > Thank you for acknowledging our clarification of issues in the rebuttal. As mentioned in Q6, based on your valuable feedback, we will revise the presentation in the paper's method section of the version to clarify our method better. We also appreciate your positive evaluation of this paper and your endorsement of our approach.
> > >
> > > As mentioned in the **global response Q1&Q2**, we will continue exploring broader applications of our method. We expect it can make contributions to the community. We also hope that the explorations we've made in our response can better demonstrate the effectiveness of RIVAL to you.
> > >
> > > Thank you again for your review and feedback on our response!

---

### Official Review · Reviewer_aMYr · 2023-07-04

**Soundness:** 2 fair
**Presentation:** 3 good
**Contribution:** 3 good
**Rating:** 7
**Confidence:** 2

**Summary:**

The authors propose a tunning-free pipeline called RIVAL(Real-world Image Variation by Alignment) for generating diverse and high-quality variations of real-world images.

In previous works, some models also generate images with novel concepts and styles but require additional training stages and data, and others directly incorporated images as the input condition results in suboptimal visual quality and content diversity.

To tackle this issue, the authors suggest a pipeline that maintains the style and semantic content of the reference without additional training stages and data.

The proposed method comprises the inverted latent chain alignment(step-wise latent normalization) and cross-image self-attention injection.

The authors determine whether aligning the inverted latent chain and injecting the modified self-attention key and value by adopting $t_{align}$ and $t_{early}$.

Unlike the other papers that use the cross-image self-attention injection method with an additional forward to calculate the cross-attention mask, the suggested approach only needs a single forward.

The authors show that RIVAL enhances performance by applying the alignment process to other text-to-image tasks through experiments and ablation studies.

**Strengths:**

- The authors claimed a novel plug-and-play pipeline to generate high-quality, real-world image variations without additional optimization through an inverted latent chain alignment and cross-image self-attention injection.
- The proposed pipeline enhances the performance of text-to-image models, such as text-driven image generation with real-image conditions and example-based inpainting. The pipeline generates images fast and flexibly with diverse text prompts adopting a new self-attention injection approach with a single forward pass.
- Compared to CFG, the advantage of the proposed pipeline is its ability to avoid misalignment between latents in two chains and its direct utilization. The authors decouple the two inference chains and rescale noise prediction during denoising inference. Furthermore, they formulate noise prediction as an adaptive normalization.
- The quantitative results outperform the other pipelines provided in the paper except for one metric, and the qualitative results show that RIVAL generates images with diverse text inputs while preserving the reference style.

**Weaknesses:**

The experiments need to be more extensive.

- The authors proposed a plug-and-play pipeline, but there are no quantitative results compared with the other plug-and-play pipelines mentioned in the paper. RIVAL used a similar scheme as MasaCtrl[2] and Plug-and-Play[3] except the latents alignment. The authors employed similar schemes to the pipeline, so comparing the quantitative results with the others would be beneficial.
- In the appendix, the proposed method allows fast and flexible text-to-image generation, but there are no results about how fast it is. It would be reasonable to compare quantitatively with other plug-and-play pipelines.

The authors need to validate the ability of the pipeline to apply to other diffusion-based generation tasks.

- The authors show the qualitative results of self-example image inpainting in Fig. 6, but the results lack experiments. It would be hard to explain the ability to extend RIVAL’s framework to other image editing tasks.

(Minor) The notation of $t_{align}$ and $t_{early}$ is confusing.

[1] Robin Rombach, Andreas Blattmann, Dominik Lorenz, Patrick Esser, and Björn Ommer. High-resolution image synthesis with latent diffusion models. In *CVPR*, pages 10684–10695, 2022.[2] Mingdeng Cao, Xintao Wang, Zhongang Qi, Ying Shan, Xiaohu Qie, and Yinqiang Zheng. Masactrl: Tuning-free mutual self-attention control for consistent image synthesis and editing. *arXiv preprint arXiv:2304.08465*, 2023.[3] Narek Tumanyan, Michal Geyer, Shai Bagon, and Tali Dekel. Plug-and-play diffusion features for text-driven image-to-image translation. *arXiv preprint arXiv:2211.12572*, 2022.

**Questions:**

- In Table 1, I wonder if using specific $t_{align}$ and $t_{early}$ values for each metric would lead to better quantitative results.
- In Fig. 6, the inpainting results show a different masked area. Can you show me the same results by overlaying the masked area at the source image like SD[1] in the paper?
- Can you provide examples of other tasks to which RIVAL can be applied?

[1] Robin Rombach, Andreas Blattmann, Dominik Lorenz, Patrick Esser, and Björn Ommer. High-resolutionimage synthesis with latent diffusion models. In *CVPR*, pages 10684–10695, 2022.

**Limitations:**

The paper needs more extensive experiments and analysis of the limitations of the proposed method.

---

> ### Author Rebuttal · Authors · 2023-08-09
>
> Dear reviewer aMYr,
>
> Thank you for your valuable feedback and constructive comments. We will address your comments below and in the revised paper.
> ### Q1 Comprehensive Quantitative Experiments
> We agree that comprehensive quantitative evaluations are vital to support our claims. Please refer to **rebuttal global@Q3** for all quantitative experiments conducted in this rebuttal.
>
> ### Q2 Comparison with Other Plug-and-Play Pipelines
> Initially, we excluded quantitative comparisons with training-free pipelines such as PnP and MasaCtrl due to task and input disparities. Specifically, PnP utilizes text to modify image appearance, while MasaCtrl leverages text for structural alterations. We recognize the merit of such a comparison in real-image editing. Please see **rebuttal Fig. G** for qualitative comparisons and the subsequent table for a quantitative evaluation.
>
> |Method|PnP|MasaCtrl|RIVAL|
> |-|-|-|-|
> |Image Alignment $\uparrow$|0.786|0.827|**0.831**|
> |Text Alignment $\uparrow$|**0.249**|0.226|0.231|
> |Palatte Distance $\downarrow$|1.803|1.308|**1.192**|
> |LPIPS $\downarrow$ |**0.245**|0.274|**0.245**|
>
> In the experiment, we directly use the inverted latent $X_{G}^{T} = X_{R}^{T}$ for two chains. We adopt the same interaction starting step $t=45$ as the MasaCtrl. Furthermore, as shown in our Appendix Fig. 14, RIVAL can create free-form content that maintains consistency with the image's style, indicating its distinction and effectiveness compared to these methods.
>
> ### Q3 Speed of the Pipeline
> As mentioned in L181-182, RIVAL processes images in roughly 8 seconds each on a single RTX 4090. When tested on the more prevalent RTX 3090, our comparisons with other plug-and-play methods confirm RIVAL's competitive inference speed against recent techniques.
>
> |Comparison on RTX 3090|PnP|MasaCtrl|RIVAL|
> |-|-|-|-|
> |Preparation Time (s)|200|6|6|
> |Inference Time (s)|21|15|15|
>
> PnP has a long preparation time due to the feature storing and long-step inversion. It's worth noting that both MasaCtrl and RIVAL can be sped up using half-precision and xformers, achieving an inference time of 6 seconds on RTX 3090. For our comparison, we used the official implementations and maintained consistent conditions across the board.
> ### Q4 Applicability to Other Tasks
> RIVAL's versatility is showcased through diverse experiments spanning multiple diffusion frameworks and applications, such as DreamBooth (Fig.9) and ImgVar[UnCLIP] (Appendix Fig.12).
>
> Additionally, we have successfully extended RIVAL's application to other tasks and frameworks, which showcases the method's adaptability and broad applicability across diverse diffusion frameworks.
>
> 1. Larger diffusion models. We've integrated RIVAL with the recent SDXL in **rebuttal Fig. A** for high-resolution (1024 $\times$ 1024) image variation and text-driven generation.
> 2. Image-conditioned controllable generation. As demonstrated in **rebuttal Fig. B**, RIVAL, in conjunction with ControlNet, produces style-consistent images under varying control conditions.
> 3. Image style transfer. Utilizing the aligned inverted latent of the content image, $X_G^T = \text{AdaIN}(X_G^T, X_R^T)$. The style transfer results are shown in **rebuttal Fig. C, the first row**.
> 4. Image editing. We have discussed it in Q2 with visuals in **rebuttal Fig. G**.
>
> ### Q5 Use of Specific $t_{align}, t_{early}$
> Your statement is correct. As stated in the grid-wise ablation in Appendix Fig.15, using complete attention feature fusion (replacement) ($t_{align} = 0$) makes the generation closely resemble the original image. However, there is a balance between two conditions (text-alignment and image-alignment). Besides, low-level texture-pasting artifacts will present when $t_{align}$ is small, as shown in Fig. 8, 4th image. The choice of $t_{early}$ primarily influences the generated images' style (color) bias.
>
> |($t_{align}, t_{early}$)|(30, 30)|(0, 30)|(30, 0)|(0, 0)|(30, 50)|(50, 30)|(50, 50)|
> |-|-|-|-|-|-|-|-|
> |Text Alignment  |0.268|0.261|0.269|0.259|0.267|_0.274_|**0.279**|
> |Image Alignment |0.846|**0.873**|0.838|_0.865_|0.839|0.813|0.817|
> |Palette Distance|1.810|**1.421**|1.806|_1.483_|2.359|2.061|2.419|
> ### Q6 Inpainting Results
> In **rebuttal Fig. E**, we've added a mask boundary overlay to highlight the generation area during inpainting. As shown in the last column, we can achieve varied generation results within the masked area by resampling the latent.
>
> RIVAL primarily aims to generate images resembling a reference. Inpainting is just one specific application of RIVAL. Given that self-example inpainting focuses on variation generation within a masked image area, it's challenging to fairly compare it with general inpainting (as RIVAL utilizes the original image as a reference) or example-based inpainting (where RIVAL is only capable when the image itself is the reference). Instead, we've offered a quantitative comparison for the image editing task in Q2.
>
> ### Q7 Limitations Analysis
> For clarity, we will include a more detailed analysis in the revised paper, using Fig. 10 examples here.
> > Semantic Bias: For a prompt like "Pokemon", bias in training set towards the popular"Pikachu" can skew generated images towards Pikachu-like designs.
>
> > Complex Scene & Hard Concepts: Stable Diffusion struggles to generate complex scenes and complicated concepts, e.g., "illustration of a little boy standing in front of a list of doors with butterflies around them". This complexity can lower inversion chain quality and widen the domain gap, leading to less accurate generation results.

---

> > ### Comment · Reviewer_aMYr · 2023-08-18
> >
> > Thanks for your responses to each question. My concerns and questions are resolved well with your rebuttal. I am willing to raise my score to accept.

---

> > > ### Author Response · Authors · 2023-08-18
> > > **Thank You for Recognizing RIVAL**
> > >
> > > Dear Reviewer aMYr,
> > >
> > > Thank you very much for recognizing our work and rebuttal. Especially in exhibiting the applicability to other tasks and comprehensive quantitative comparisons, your questions have helped us engage in a more comprehensive discussion and comparison of RIVAL's mechanism.
> > >
> > > We will integrate the pertinent discussions into the revision. Thank you again for your review and feedback on our response!

---

### Official Review · Reviewer_mFBx · 2023-07-23

**Soundness:** 3 good
**Presentation:** 3 good
**Contribution:** 3 good
**Rating:** 7
**Confidence:** 5

**Summary:**

Generating real-world image variations is an important research task with practical applications such as image editing, image synthesis, and data augmentation. Past approaches include texture synthesis, neural style transfer, and generative models, among others. This study proposes a method called "Real-world Image Variation by ALignment", which generates high-quality image variations from a single image sample through the inference process of an aligned diffusion model. The paper provides experiments on different images.

**Strengths:**

The overall effect is very good. The proposed method takes full advantage of the properties of the diffusion model.



**Weaknesses:**

There are a lot of recent papers based on a very well trained diffusion generative model. The results of these papers are generally very good (after all, it is based on a very mature large model). These papers generally have some micro-designs, which make the academic meaning of the papers more prominent. However, such papers have a common problem, that is, they do not discuss the research issues in a very specific way. Often only stay in the effect. But this hardly accounts for the actual scholarly progress of these papers. Chances are it's just a manifestation of the capabilities of the base model itself.

Taking this paper as an example, the paper proposes the method of Aligning Diffusion Inversion Chain. But the necessity and interpretability of the method is not fully discussed. In fact, there are many "tricky" ways to achieve the effect of this paper, and it is not necessary to go through the method described in Section 3 (By the way, the writing of Section 3 is very difficult to understand. In fact, a lot of useless mathematics is unnecessary. It is recommended that authors use more carefully set diagrams to explain their methods). I doubt the necessity of the proposed method. I hope the author can have a full ablation experiment, and compare and discuss it with a wide range of existing tricks. Of course, I also realize that such a requirement is a bit too harsh. Because many methods and tricks have not been written into papers or formally disseminated. The authors may not be able to adequately discuss every possibility.

Interpretability is also another important topic. I want to know the specific physical meaning of each latent and representation in the inversion transfer or the point that it can be exploited in this method. This paper doesn't really explain it (or even try to).

=====================

Post rebuttal:

Thanks for the rebuttal. I am willing to rase my score to accept (7). But I recommend authors to add these new discussions in to the final version.

**Questions:**

Two general questions:
1. Ablation study.
2. Interpretability.

Please see Weaknesses for full details.

**Limitations:**

Yes

---

> ### Author Rebuttal · Authors · 2023-08-09
>
> Dear reviewer mFBx,
>
> Thank you for your valuable feedback and constructive comments. We will address your comments below and in the revised paper.
> ### Q1 Utilizing Capability of the Base Model
> We respect your comments on the trend of recent works leveraging mature models. While we agree these base models are crucial, RIVAL  innovates by addressing the specific challenge of controllable, high-quality, real-world image generation. RIVAL is designed to extend beyond the capabilities of the original model and can be generically applied to broad well-trained diffusion frameworks.
>
> This generalization ability is demonstrated through extensive experiments with diverse diffusion frameworks and applications, including DreamBooth (Fig.9), SDXL (**rebuttal Fig.A**), ControlNet (**rebuttal Fig.B**), and ImgVar (Appendix Fig.12). We posit that this breadth of applicability constitutes scholarly progress, broadening the base model's utility.
> ### Q2 Clarification of Section 3
> For a detailed and precise workflow of our method, please refer to the Pytorch-style pseudocode for RIVAL in **SLMY@Q3**. A LaTeX algorithm version will also be included in the revision.
> ### Q3 Necessity of RIVAL: Wider Discussion and Comparisons
> To elucidate RIVAL's necessity, we delve into two key aspects:
> #### Q3-1 Module-wise Ablations
> Agreeing with your suggestion for thorough ablations, we augment our visual ablations (**rebuttal Fig. F**, Fig.8 and Appendix Fig.15-17) with a comprehensive module-wise experiment.
> |Module|Attention Inject|Early Fusion(Eq.3)|Latent Init.(Eq.8)|Noise Align(Eq.9-10)|Palette Dist.|Text Align|Image Align|
> |-|-|-|-|-|-|-|-|
> |Baseline (SD)|||||3.917|0.266|0.751|
> |Attn Inject|Y||||3.564|0.277|0.804|
> |Attn Fusion|Y|Y|||3.518|0.274|0.820|
> |Latent Init.|||Y||3.102|0.268|0.764|
> |Noise Align||||Y|3.661|0.251|0.647|
> |w/o Attn Operations|||Y|Y|2.576|0.276|0.760|
> |w/o Fusion&Noise Align |Y||Y||2.419|**0.279**|0.817|
> |w/o Attn Fusion|Y||Y|Y|1.902|0.274|0.818|
> |w/o Latent Init|Y|Y||Y|3.741|0.242|0.653|
> |w/o Noise Align|Y|Y|Y||2.335|0.267|0.839|
> |**Full Model**|Y|Y|Y|Y|**1.810**|0.268|**0.846**|
>
> The results underscore the role of each component and their combinations. Attention injection facilitates high-level feature interactions for better condition alignment (both text and image). Early fusion, built upon attention injection, aids in early step chain alignment, significantly enhancing image alignment. Meanwhile, latent noise alignment guarantees the preservation of color.
>
> Latent initialization has a pronounced impact, notably enhancing the color palette metric, an effect intensified by noise alignment. A thorough evaluation of hyperparameters $t_{align}$ and $t_{early}$ can be found in **aMYr@Q5**, Appendix Fig.15, and analysis in Appendix L84-89.
> #### Q3-2 Wider Discussions and Comparisons
> To further ascertain RIVAL's necessity and effectiveness, we compare it with three concurrent related works. **Visual results: rebuttal Fig. C&G. Quantitative comparison: aMYr@Q2**.
> 1. **Tunning-based style transfer method [StyleDrop]** Similar to ours, this method yields visually appealing results with a style exemplar image. Despite its high-quality stylization, it relies on case-wise finetuning with intricate prompt design. At the same time, its base model MUSE isn't open-sourced.
> 2. **Attention-based image editing [PnP]** PnP employs a plug-and-play approach that uses attention and feature injections for structure-preserving text-based editing. While the results are good, the image style diverges from the exemplar due to a lack of chain alignment consideration.
> 3. **Attention-based image synthesis [MasaCtrl]** This work employs a similar attention-injection framework. We've comprehensively analyzed the differences in attention in Appendix Section B and Fig.11. MasaCtrl is generally suited for real-image editing instead of unconstrained generation. We present a visual comparison in **rebuttal Fig. G**. As input latent is fixed, it cannot generate free-form images from a single target prompt, as results in Appendix Fig. 14.
> ### Q4. Interpretability of RIVAL
> We add explanations and experiments for the interpretability of RIVAL in the following two aspects.
> #### Q4-1 Latent Similarity
> To assess RIVAL components' efficacy regarding the distribution gap, we illustrate the KL divergence of noisy latent between chains A and B, $X_{A}^{t}$ and $X_{B}^{t}$ in the generation process, as depicted in **rebuttal Fig. D left part**. Interactions between different distributions (green line) widen the gap, while two latent chains with the same distribution (yellow line) can get a better alignment by attention interactions. With aligned latent chain alignment and interaction, RIVAL (red line) effectively generates real-world image variations.
> #### Q4-2 Reference Feature Contribution
> Attention can be viewed as sampling value features from the key-query attention matrix. RIVAL converts self-attention to image-wise cross-attention. When latents are sourcing from the same distribution ($X_G^T, X_R^T\sim\mathcal{N}(0, I)$) images retain consistent style and content attributes  (Fig. 3, first two images, **rebuttal Fig. D left, yellow line**). This result is beneficial since we do not require complex text conditions $c$ to constrain generated images to get similar content and style.
>
> For a more direct explanation, we visualize of images' bottleneck feature contributions (as in Fig. 7 right part) of attention score are presented in **rebuttal Fig. D right part.**
> Reference contribution of the softmax score is:
> > $$\text{score}_{R}=\frac{\sum^{v_i\in\mathbf{v}_R}({W^Q\mathbf{v}_G\cdot(W^K (v_i))^{\top}})}{\sum^{v_j\in\mathbf{v}_G\oplus\mathbf{v}_R}({W^Q\mathbf{v}_G\cdot(W^K(v_j))^{\top}})}$$
> As RIVAL adopts early fusion in the early steps, we use 50% as the score in the early steps. Results indicate that RIVAL leverages 30% of the reference feature in the final step, compared to random initialization (7%) and latent initialization (21%).

---

> > ### Comment · Reviewer_mFBx · 2023-08-11
> > **Raising My Score**
> >
> > Thanks for the rebuttal. I am willing to rase my score to accept (7). But I recommend authors to add these new discussions in to the final version.

---

> > > ### Author Response · Authors · 2023-08-11
> > > **Thank You for Recognizing RIVAL**
> > >
> > > Dear Reviewer mFBx,
> > >
> > > Thank you for recognizing our work and the content of our rebuttal. The questions you raised in the review are essential for explaining the effectiveness of RIVAL and have helped us to delve deeper into the underlying mechanisms of this work.
> > >
> > > We will add all relevant experiments and discussions in the revised version. Besides, we will organize the content in the method section to make the article more understandable and academically meaningful.
> > >
> > > Thank you again for acknowledging our work!

---

### Official Review · Reviewer_4HxC · 2023-07-26

**Soundness:** 3 good
**Presentation:** 3 good
**Contribution:** 3 good
**Rating:** 6
**Confidence:** 4

**Summary:**

The paper investigates the domain gap between generated images and real-world images, as it is challenging in generating high-quality variations of real-world images. It mainly contains two contributions, the first one is the cross-image self-attention injection for generating images that correspond to the given exemplar, and the second one is the latent alignment, which can help to eliminate the biased tone and semantics. The comparison with stable diffusion, SD variation, and ELITE validates the effectiveness of each component. Lastly, the ablation studies analyze in depth the contribution of each submodule, showing the positive impact of each part.

**Strengths:**

+ The paper is well-written and easy to follow for most parts, except when introducing the details of the shuffle operation in equation 8. It is not easy to directly understand the operation.
+ The paper has a nice flow of descriptions and figures to illustrate the qualitative comparison.
+ The diagrams are easy to understand and are well-made.
+ The main contribution of the proposed cross-image self-attention injection and latent alignment are intuitive and well-executed in this paper. Each design is reasonable to handle the corresponding challenges, like object custom and style consistency.
+ The ablation study is sufficient to validate the effectiveness of each component in designing the whole network, the concatenation of cross-image features in different steps, and the alignment of two features.
+ Experiments show superior performance in comparison with recent works.

**Weaknesses:**

- Generally speaking, this work is more like the recent customized text-to-image generation, except that this work considered style consistency, like the analyses in Figure 4 (2nd-row). Since these competing methods did not consider this style factor, so they easily have the tones inconsistency problem. Directly comparing with them seems not very fair, as they can generate great results, except for the inconsistent tone distribution. In my opinion, it seems not a crucial issue.
- The concatenation of v_G and v_R in equation 3 is also like some training-free video generation, which adopts the cross-frame interaction. The second proposed latent alignment is widely used in style transfer tasks.

**Questions:**

Some deep analyses of the failed cases seem not enough. The provided examples cover very wide ranges, like semantic bias, complex scenes, and hard concepts.
What is the difference between custom image generation and variation generation? It seems that it only has the style difference, as this work needs to keep the style consistent with the given exemplar. In this case, I think the authors should discuss these related works, and compare with them by considering style transfer factors.

---

> ### Author Rebuttal · Authors · 2023-08-03
>
> Dear reviewer 4HxC,
>
> Thanks for your valuable feedback and constructive comments. We will address your comments below and in the revised paper.
> ### Q1 Shuffle Operation
>
> In Eq. 8, the shuffle operation $X_G^T=\text{shuffle}(X_R^T)$ is designed to rearrange latent elements within the spatial dimension.
>
> We present a PyTorch-style code snippet for better understanding:
> ```python
> # x_rt: latent variable with the shape (batch size, num. of dimensions, height, width)
> def shuffle(x_rt):
>     bs, n_dim = x_rt.shape[:2]
>     # Generate a permutation index
>     perm_idx = torch.randperm(x_rt.nelement() // (bs * n_dim))
>     # Reshape, permute, and reshape back to the original
>     x_gt = x_rt.view(bs, n_dim, -1)[:, :, perm_idx].view(x_rt.size())
>     return x_gt
> ```
> ### Q2 Fairness of Comparison
> We recognize the importance of ensuring a fair comparison. In the image variation task, RIVAL uses the same objectives and input states as ImgVar and ELITE, ensuring a fair comparison. Our comparisons aim to highlight the benefits of RIVAL, particularly regarding style and content consistency, an overlooked yet critical aspect in image generation.
> #### Q2-1 Comparison and Importance of Image Variation
> Image variation in diffusion-based generation stems from UnCLIP (DALLE2), using image embedding as a condition for tailored generation.
>
> In Fig. 4 and Tab. 1, the StableDiffusion baseline takes only text as a prompt to illustrate the benefits of introducing a reference image to guide the generation process. Concerning ImgVar and ELITE, these methods also use the original image as a condition and are fine-tuned on large-scale real-world datasets to align the image condition with generated images. However, they often fail to address specific subtle attributes that may not be easily described from the image condition, as shown in Fig 4 (e.g., out-of-focus background in the 3rd row, blinds in the 5th row).
>
> Such attributes define an image's unique identity. Our approach strives to generate images that closely align with the original image from random noise. This property is vital as it is the fundamental precondition of all the applications presented in the paper and rebuttal, allowing us to generate images more similar to real-world exemplars.
> #### Q2-2 Style-transfer Factor Consideration
> Beyond image variation, RIVAL's capabilities in style transfer are evident. This ability is particularly highlighted in its text-driven image generation (Fig. 5) and customization (Fig. 9). Such demonstrations further accentuate RIVAL's strengths in producing style-consistent images.
>
> Furthermore, we compare methods explicitly crafted for style transfer:  StyleDrop and InST in **rebuttal Fig. C**. This comparison underscores RIVAL's versatility, especially when considering the case-wise fine-tuning (~20min) that StyleDrop and InST demands.
> ### Q3 Novelty of Purposed Modules
> Here we will describe our ideas to answer how we can solve problems through the organic combination of modules that might appear "simple" at first glance. A comprehensive ablation is presented in **rebuttal Fig. F and mFBx@Q3-1**.
> #### Q3-1 Cross-frame Self-Attention Injection
> As you correctly pointed out, we have reviewed some video-generation methods (L110-112) that use self-attention injection for cross-frame consistency. Despite sharing a similar objective, our technique is distinct as we are among the first to adapt it to real-world image generation challenges.
>
> A primary challenge we address is the attention-attenuation (L132-135, **rebuttal Fig. D right part**). This problem arises due to the discrepancy between the distributions of latents in image generation. In video generation methods, latents across different frames generally typically maintain the same distribution, whether they originate from inverted latents by DDIM [Video-P2P, Fate-ZeRO] or are derived from the standard Gaussian [Tune-A-Video, Gen-1]. However, uniformity cannot be presumed for generation starting from random noise and referencing with an inverted latent.
>
> Although DDIM inversion is deterministic, it does not ensure that the inverted latent adheres to the standard distribution $\mathcal{N}(0, I)$. This discrepancy in the distribution between the real-inversion and generation chains becomes evident when applying cross-image self-attention (Sec. 3.1). This results in a distribution gap in the Key-Query-Value (KQV) terms. This gap can impact attention computation, leading to varied generation results. Hence, our attention-fusing design combined with latent alignment is proposed to address this challenge effectively.
> #### Q3-2 Latent Noise Alignment
> While adaptive normalization is commonly observed in style-transfer and domain adaptation tasks [StyleGAN, AdaIN], our approach to latent alignment tackles a distinct challenge inherent to diffusion generation. Specifically, we aim to solve realism degradation from high classifier-free guidance (CFG).
>
> High CFG produces text-aligned images but can cause realism-divergent artifacts **(rebuttal Fig. H)**. To counter this, we introduced a decoupled adaptive normalization (Eq.9-10). It decouples CFG generation from DDIM inversion in one pass, aligning latents to reflect the image exemplar faithfully.
> ### Q4 Limitation Analysis
> For clarity, we will include a more detailed analysis in the revised paper using Fig. 10 examples here.
> > Semantic Bias: A prompt like "Pokemon" may lean towards popular choices like "Pikachu" due to training set biases, resulting in a Pikachu-dominated generation.
>
> > Complex Scene & Hard Concepts: Stable Diffusion struggles to generate complex scenes and complicated concepts, e.g., "illustration of a little boy standing in front of a list of doors with butterflies around them in Fig. 10(b)". This complexity can degrade the inversion chain and widen the domain gap, leading to less accurate results.

---

> > ### Comment · Reviewer_4HxC · 2023-08-16
> >
> > I appreciate the authors' detailed response and my concerns are addressed well. Considering other reviews, I would like to give weak accept.

---

> > > ### Author Response · Authors · 2023-08-16
> > > **Thank You for Recognizing RIVAL**
> > >
> > > Dear Reviewer 4HxC,
> > >
> > > Thank you very much for recognizing our work and the effort we put into our rebuttal. Especially in explaining this paper's novelty, your questions have helped us engage in a more comprehensive discussion and comparison of RIVAL's mechanism.
> > >
> > > We will integrate the pertinent discussions, especially concerning the comparison's fairness and the method's novelty, into the subsequent revision.

---

### Author Rebuttal · Authors · 2023-08-09

Dear all reviewers,

We want to express our gratitude to all reviewers for their thorough reviews and constructive comments. In the global rebuttal, we aim to summarize and address representative questions raised by the reviewers.

**We encourage all reviewers to refer to the global response for Q1 to gain a deeper insight into our motivation and applications.**
### Q1 Motivation and Problem Setting [all reviewers]
The primary goal of our paper is to introduce a tuning-free approach to produce images that resemble real-world samples. With a traditional T2I (Text-to-Image) pipeline, one can generate an image in the "Starry Night" style simply by inputting the prompt "in Starry Night style." However, considering that Van Gogh created 864 oil paintings in his lifetime using various techniques, generating images that closely align with an arbitrary real-world reference remains an essential problem.

While our objective may seem similar to image stylization, a fundamental distinction exists between generation (conditioned on image) and tasks based on the original image (like image stylization and editing). While generation often starts from random noise and produces free-form, unconstrained images, our method can act as a style-condtioned generation technique when references incorporate low-level features. Here, we initialize the latent as a normalized inverted latent of the content image. This process also allows us to harness image editing capabilities directly using the inverted latent.

To provide clarity, we present a table comparing various diffusion-based methods based on four key aspects: (text prompt, image exemplar, initial latent, and requires finetuning)
|Application |Text Prompt|Image Exemplar|Gen. from noise|Gen. from inverted latent|Requires Tuning|Representative Method|
|-|-|-|-|-|-|-|
|T2I generation|Y|-|Y|-|-|Stable Diffusion|
|Image Variation*|C|Y|Y|-|-|DALLE2|
|Style conditioned T2I*|Y|Y|Y|-|Y|StyleDrop|
|Image Stylization**|C|Y|-|Y|Y|InST|
|Image Editing**|Y|Y|-|Y|-|Prompt2Prompt, MasaCtrl|
|Image Customization|C|Y|C|-|C|DreamBooth, Custom Diffusion|
|**Our Method**|Y|Y|C|C|-|**RIVAL**|

>**Y: required, C: required in some methods/application**\
*: Demonstrated ability of RIVAL in the paper submission.\
**: Demonstrated ability of RIVAL in the rebuttal.

### Q2 More Applications [4HxC, aMYr, 5k69]
Given that RIVAL is a broad-based idea for the diffusion model during denoising inference, we received suggestions to expand our method to other diffusion-based applications. Below is a list of tasks RIVAL is adept at:
|Applications of RIVAL |Text Prompt|Image Exemplar|Gen. from noise|Gen. from inverted latent|
|-|-|-|-|-|
|Image Variation|Y|Y|Y|-|
|Style conditioned T2I|Y|Y|Y|-|
|Structure-guided T2I (ControlNet)|Y|Y (+Control condition)|Y|-|
|Image Stylization|Y|Y|-|Y|
|Image Editing|Y|Y|-|Y|

Building on Q1, we have expanded our approach to cover the three new applications listed in the last three columns for adapting RIVAL to different tasks. Generally, RIVAL is suitable for tasks that need consistent style/content generation in line with the exemplar.

### Q3 Table of Content: Experiments [mFBx, aMYr, 5k69]
Outlined below are the quantitative experiments conducted in this rebuttal:

1. **[mFBx@Q3-1] Module-wise ablation study**: We present a detailed module-wise ablation study to evaluate the impact of our method's four components (attention injection, attention feature fusion, latent initialization, and noise alignment).

2. **[aMYr@Q2] Comparison with plug-and-play methods**: We compare our image editing outcomes with PnP Diffusion and MasaCtrl.

3. **[aMYr@Q3] Speed comparison**: We delve into a time comparison for preparation and inference phases against PnP and MasaCtrl, with RIVAL showing competitive inference speed.

4. **[aMYr@Q5, 5k69@Q2] Hyperparameter Ablation**: Quantitative results are shown for selected timestep hyperparameters controlling the denoising process and for different CFG values, proving RIVAL's capability to mitigate artifacts from high CFG values.

### Q4 Table of Content: Visualizations [all reviewers]
We have made available a PDF containing visualization results of all pertinent experiments. We suggest reviewers examine our results, especially Fig. A, B, C, G, and H, within the provided PDF for comprehensive details.
### Q5 Clarifications
#### Q5-1 The algorithm workflow [mFBx, SLMY]
We appreciate the feedback and admit that our paper's presentation could be more precise. We have added an algorithmic description in **SLMY@Q3** to help reviewers better understand RIVAL's workflow.
#### Q5-2 Limitation analysis [4HxC, aMYr]
This is presented in **4HxC@Q7 and aMYr@Q7**.

#### Q5-3 Correction of formula expressions
Despite no direct feedback, we have spotted and rectified a typo in our inversion process in Eq. (5-6). Take Eq. (5) as an example. A coefficient $\sqrt{\alpha_{t-1}}$ should be multiplied before $(\beta_{t-1} - \beta_{t})$,
$$X^{t-1}=\sqrt{{\alpha_{t-1}}/{\alpha_t}} \cdot X^t+\sqrt{\alpha_{t-1}}* (\beta_{t-1} - \beta_{t}) \cdot \varepsilon_\theta(X^t, t, \mathcal{C}).$$

Furthermore, a simpler and corrected version of the adaptive normalization in Eq.(9) is:
$$\epsilon_\theta(X_G^{t}, t, \mathcal{C}) = \text{AdaIN}(\varepsilon_\theta^\mathrm{cfg}(X_G^{t}, t, \mathcal{C}), \varepsilon_\theta(X_R^{t}, t, \mathcal{C}))\text{, where}$$
$$\text{AdaIN}(a, b) = \sigma(b) (\frac{a - \mu(a)}{\sigma(a)}) + \mu(b).$$

In addition to the existing references, we have incorporated a few new ones in this rebuttal, which will appear in the revised version.

> *Inversion-Based Style Transfer With Diffusion Models*; **Yuxin Zhang et al.**; Proceedings of the IEEE/CVF Conference on Computer Vision and Pattern Recognition (CVPR), 2023, pp. 10146-10156

> *StyleDrop: Text-To-Image Generation in Any Style*; **Kihyuk Sohn et al.**; arXiv 2023

> *SDXL: Improving Latent Diffusion Models for High-Resolution Image Synthesis*; **Dustin Podell et al.**; arXiv 2023

---

### Decision · Program_Chairs · 2023-09-21

**Decision:**

Accept (spotlight)

**Comment:**

All reviewers are positive on the paper, recognizing its technical novelty and potential impact. The AC sees no reason to interfere with reviewer consensus, and is happy to recommend acceptance.